# BARREL: Boundary-Aware Reasoning for Factual and Reliable LRMs

**Junxiao Yang**[1,2] , **Jinzhe Tu**[1], **Haoran Liu**[1], **Xiaoce Wang**[1], **Chujie Zheng**[1],
**Zhexin Zhang**[1], **Shiyao Cui**[1], **Caishun Chen**[2], **Tiantian He**[2], **Hongning Wang**[1],
**Yew-Soon Ong**[2,3], **Minlie Huang**[1]*

[1] The Conversational AI (CoAI) group, DCST, Tsinghua University
[2] Centre for Frontier AI Research, Institute of High Performance Computing,
Agency for Science, Technology and Research, Singapore
[3] The College of Computing and Data Science, Nanyang Technological University
yangjunx21@gmail.com, aihuang@tsinghua.edu.cn

## Abstract

Recent advances in Large Reasoning Models (LRMs) have shown impressive capabilities in mathematical and logical reasoning. However, current LRMs rarely admit ignorance or respond with "I don't know". Instead, they often produce incorrect answers while showing undue confidence, raising concerns about their factual reliability. In this work, we identify two pathological reasoning patterns characterized by overthinking that contribute to the overconfident and incorrect answers: *last-minute guessing* and *second-thought spiraling*. To address these issues, we propose BARREL—a novel framework that promotes concise and boundary-aware factual reasoning. Our experiments show that BARREL-training increases the reliability of DeepSeek-R1-Distill-Llama-8B from 39.33% to 61.48%, while still achieving accuracy comparable to models finetuned on reasoning data generated by R1. These results demonstrate that our pilot study is inspiring to build more reliable and factual *System 2* LRMs.

## 1 Introduction

Recent advances in Large Reasoning Models (LRMs) (Jaech et al., 2024; Guo et al., 2025; Team, 2025) have shown impressive performance in specialized reasoning tasks, especially in mathematics and logic. However, these gains have not led to corresponding improvements in reliability. On the contrary, faithfulness hallucination rates are rising (Hughes & Bae, 2023), and helpfulness on factual tasks is declining (Zhao et al., 2025), raising concerns about the reliability of these reasoning models.

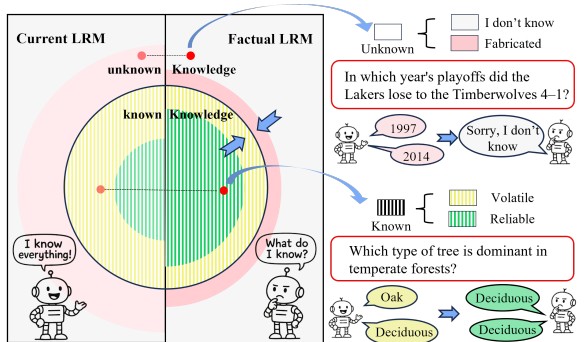

Figure 1: How reliable factual reasoning is expected to improve model performance. **Left:** Current LRMs rarely admit ignorance and often respond inconsistently. **Right:** Reliable LRMs should acknowledge unknowns and express known facts more consistently.

In this work, we focus primarily on the factual reliability of LRMs, which is a crucial requirement for many real-world tasks. Factuality of language models involves two aspects: *knowing* (whether the model holds relevant knowledge) (Huang et al., 2025; Ji et al., 2023) and *telling* (conveying the correct factual information) (Gekhman et al., 2024; Mallen et al., 2022). As shown in Figure 1, we want LRMs to exhibit two aspects of factual reliability: (1) Identify both what it knows and what it does not know: The model should be able to recognize questions it does not know the answer and respond with "I don't know." (Xu et al., 2024a; Zhang et al., 2024a) (2) Tell what it knows: There is a gap

---

*Corresponding author.

between knowing and telling (Saunders et al., 2022), and we want to improve the model's accuracy in expressing the knowledge it has (Zhang et al., 2024b).

However, current LRMs consistently struggle with the two factual reliability goals above. They rarely acknowledge gaps in their knowledge and often fabricate answers instead, even on questions for which they lack sufficient knowledge. Moreover, their responses can be inconsistent—providing incorrect answers in some instances while correctly responding to similar queries elsewhere (Wang et al., 2022), thereby reducing the overall factual accuracy of their responses (Zhao et al., 2025).

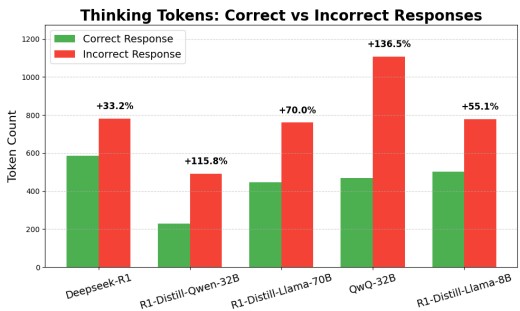

Figure 2: Number of reasoning tokens used by LRMs when producing correct versus incorrect answers. We test on TruthfulQA across different types of reasoning models. Details and results on other datasets are listed at Appendix A.

What underlying reasoning behaviors contribute to these observed issues? As shown in Figure 2, our preliminary experiments reveal a variant of the overthinking pattern discussed by Chen et al. (2024), which we term *factual overthinking*—where large reasoning models (LRMs) consume more tokens when producing factually incorrect answers than when generating correct ones. And we further identify two prevalent pathological reasoning patterns associated with this overthinking phenomenon. As shown on the left part of Figure 3, one prominent pattern is ***Last-minute Guessing***, in which the model, after extensive but inconclusive reasoning, abruptly commits to an answer in a final burst of speculative output—much like a student hastily filling in an answer just before the end of an exam. Another frequent pattern is ***Second-thought Spiraling***, where the model initially identifies the correct answer but continues to over-analyze, ultimately undermining its own correct conclusion.

To mitigate these pathological reasoning patterns, we propose a novel training framework, **BARREL** (Boundary-Aware Reasoning for Reliable and Factual LRMs). As shown in Figure 3, BARREL trains LRMs to perform concise, deliberative factual reasoning and draw conclusions after it has explored a sufficient number of candidates. Concretely, for known questions, when the model identifies the correct factual answer during reasoning, we want it to maintain confidence in that answer while continuing to explore other possible ones. Once there are no likely candidates, it should halt further reasoning and provide the correct factual answer. For unknown questions, after exploring a sufficient number of plausible candidates, it should terminate reasoning and proactively admit its lack of knowledge with uncertainty-aware refusal (e.g., "Sorry, I don't know"). To implement BARREL, we begin by employing a sampling strategy to probe the model's knowledge boundary. Drawing on the identified pathological patterns, we construct two distinct types of reasoning data, and use Supervised Fine-Tuning (SFT) to instill the corresponding reasoning behaviors in the model. Finally, we adopt Group Relative Policy Optimization (GRPO) using general reliability-based reward (high for correct answers, medium for uncertainty-aware refusal, and low for incorrect answers) without the need for labeling *known*/*unknown*, further enhancing the model's ability to generalize in factual reasoning.

Our experiments demonstrate that models trained with BARREL can effectively express uncertainty-aware refusal, and mitigate the two pathological reasoning patterns. This capability significantly improves reliability: BARREL boosts the reliability of DeepSeek-R1-Distill-Llama-8B from 39.33% to 61.48%, while maintaining an accuracy of 40.7%, which is even higher than the accuracy of 38.43% achieved by distillation. Through detailed analysis, we highlight the critical role of medium-level rewards in promoting uncertainty-aware refusal. This result also identifies the root cause of models' inability to admit ignorance to a fundamental gap in current RL paradigms: they do not reward refusal. As a result, models are incentivized to answer every question, regardless of uncertainty.

Our main contributions are as follows:

- We discover the factual overthinking phenomenon and point out two pathological reasoning patterns that lead to factual unreliability of LRMs.
- To the best of our knowledge, we are the first to explore how LRMs can utilize reasoning to admit ignorance and say "I don't know". Also, we introduce a novel training pipeline to do so.
- We emphasize medium-level rewards to encourage uncertainty-aware refusal for reliable LRMs

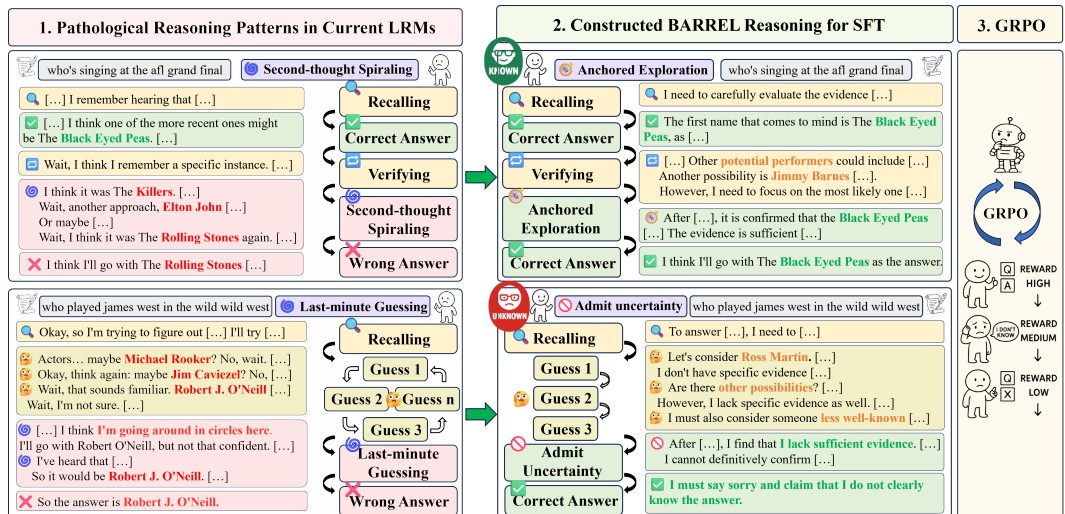

Figure 3: **Left:** The two current reasoning patterns of LRM: Last-minute Guessing, typically associated with unknown knowledge, and Second-thought Spiraling, which occurs despite known knowledge. **Right:** The BARREL pipeline addresses both cases by correcting overthinking tendencies and constructing SFT data accordingly, further enhanced with GRPO.

## 2 RELATED WORK

**Knowledge Boundary** The knowledge boundary of LLMs refers to the extent of knowledge a model possesses or can reliably recall (Li et al., 2024). Identifying such boundaries is crucial for model safety and reliability and is commonly addressed by confidence calibration (Ren et al., 2023), internal state probing (Ji et al., 2024), uncertainty estimation (Kapoor et al., 2024), and sampling strategy (Gekhman et al., 2024; Xue et al., 2024). To mitigate failures by outbound queries, recent studies have proposed training LLMs to abstain from answering when uncertain, often by generating "I don't know" responses (Cheng et al., 2024) or providing further explanations of the unanswerability (Deng et al., 2024). Our work pushes it further to structured and interpretable rationales about their knowledge boundary.

**Factual Alignment** Factual alignment aims to improve factuality while preserving instruction-following capabilities in LLMs. Recent work incorporates factuality-awareness into both SFT and RL stages to improve factual accuracy (Lin et al., 2024), or finetuning with DPO to enhance its self-evaluation capability (Zhang et al., 2024b). Other approaches include fine-tuning with refusal-aware datasets to encourage appropriate abstention behaviors (Zhang et al., 2024a), and RLKF, which guides models to reject uncertain queries based on external feedback signals (Xu et al., 2024a). Similarly, other reinforcement learning strategies have been proposed to incentivize models to express doubt or uncertainty (Stangel et al., 2025; Xu et al., 2024b). Damani et al. (2025) has also explored training LLMs to reason with uncertainty using binary rewards. While previous work has primarily focused on non-reasoning models, our research demonstrates how to correct reasoning pathologies to enhance the factual reliability of LRMs.

## 3 METHOD

In this section, we introduce the overall framework of BARREL, which comprises three main components: (1) **Knowledge Labeling**, which distinguishes whether a question is known or unknown to the model; (2) **Reasoning Trace Construction for SFT**, which constructs reasoning traces based on the question type and performs SFT to prepare the model to follow this thinking pattern; (3) **GRPO Stage**, which further enhances the model using a rule-based factual reward. We detail each component below.

## 3.1 KNOWLEDGE LABELING

We first determine whether a question is known to the target model using the sampling strategy proposed in (Gekhman et al., 2024), which is a widely recognized approach (Xue et al., 2024; Li et al., 2024). Let $\mathcal{D} = \big\{(x_i, y_i^*)\big\}_{i=1}^{N}$ be a factual QA dataset, where each question $x_i$ has a ground-truth answer $y_i^*$. We generate answers with the target model $\mathcal{M}$ using $K$ distinct few-shot prompts $\{\mathcal{P}_j\}_{j=1}^{K}$ and repeat the sampling procedure $L$ times for every prompt:

$$y_i^{j,k} \sim \mathcal{M}(\cdot \mid \mathcal{P}_j \,\|\, x_i), \qquad j = 1, \ldots, K, \ k = 1, \ldots, L. \tag{1}$$

After collecting the samples $\mathcal{Y}_i = \big\{\, y_i^{j,k} \big\}_{j=1,k=1}^{K,\,L}$, we consider a question *known* to the model if at least one sampled answer matches the ground-truth answer under evaluator $E$.

$$l_i = \begin{cases} \textit{known}, & \text{if } \exists\, y \in \mathcal{Y}_i \text{ such that } E(y, y_i^*) = 1, \\ \textit{unknown}, & \text{otherwise.} \end{cases} \tag{2}$$

## 3.2 REASONING TRACE CONSTRUCTION FOR SFT

To address the pathological reasoning patterns identified in our analysis—namely, Last-minute Guessing and Second-thought Spiraling—we propose a targeted method for constructing reasoning trajectories. Based on the type of question, we construct two distinct evidence-grounded reasoning traces $\mathcal{T}(x_i)$ for a question $x_i$, aiming to respectively correct these two faulty reasoning patterns. This construction is outlined in Algorithm 1.

To mitigate Second-thought Spiraling in **known** questions, where the gold answer $y^*$ with strong evidence $e^*$ is available, it should begin by retrieving and identifying this answer. It then examines alternative candidates $(y_j, e_j)$ to contrast possibilities. After this anchored exploration, it reaffirms the choice with solid justification and draws a confident conclusion favoring the correct answer.

To address Last-minute Guessing in **unknown** questions, the system adopts a similar exploratory strategy: it recalls background knowledge and searches on plausible answer-evidence pairs $(y_j, e_j)$ through hypothesizing. However, if it fails to identify a sufficiently supported answer, it explicitly acknowledges the uncertainty and ultimately outputs a cautious, confirmed rejection—demonstrating its ability to explore high-probability paths without overcommitting or hallucinating.

---

**Algorithm 1** BARREL reasoning trace $\mathcal{T}(x_i)$ construction

---

    **Input:** Question $x_i$ and knowledge label $l_i$, gold answer with evidence $(y^*, e^*)$, alternative candidates with poor evidence $\{(y_j, e_j)\}_{j=1}^{n}$
    **Output:** reasoning trace $\mathcal{T}(x_i)$
1:  $\mathcal{T}(x_i) \leftarrow \langle\rangle$                                           ▷ Initialize an empty trace
2:  $\mathcal{T}(x_i) \mathrel{+}= \text{RECALL}(x_i)$                  ▷ record recalled background facts
3: **if** $l_i = 1\,(known\ knowledge)$ **then**
4:     $\mathcal{T}(x_i) \mathrel{+}= \langle y^*, e^* \rangle$           ▷ Attach gold answer and supporting evidence
5:     $\mathcal{T}(x_i) \mathrel{+}= \{(y_j, e_j)\}_{j=1}^{n}$         ▷ Attach distractor answer-evidence pairs
6:     $\mathcal{T}(x_i) \mathrel{+}= \text{CONFIRM}(y^*)$     ▷ Verify the conclusion with strong evidence
7: **else** $(unknown\ knowledge)$
8:     $\mathcal{T}(x_i) \mathrel{+}= \{(y_j, e_j)\}_{j=1}^{n}$         ▷ exploring plausible answer-evidence pairs
9:     $\mathcal{T}(x_i) \mathrel{+}= \text{Acknowledge Uncertainty}()$     ▷ Record uncertainty for guesses
10: **end if**
11: **return** $\mathcal{T}(x_i)$                      ▷ Return the constructed reasoning trace

---

We construct the reasoning traces by prompting GPT-4 with detailed instructions and BARREL reasoning examples. This approach produces a Long-CoT-style reasoning process that aligns with the expected reasoning patterns. Examples of the constructed reasoning traces for both known and unknown questions are shown in Figure 3, and the detailed prompt used for trace construction is provided in Appendix G.

Then, we use these data to train the model to emulate boundary-aware and deliberative reasoning patterns using SFT. For each question $x_i$, we construct full output $o_i^* = \mathcal{T}(x_i)\|a_i$, where $a_i$ is

either the gold answer $y_i^*$ (for known questions) or an uncertainty-aware refusal (e.g., "Sorry, I don't know") for unknown questions. This instills the model with a disciplined reasoning style grounded in traceable evidence and uncertainty-aware conclusions. The training objective minimizes the negative log-likelihood:

$$\mathcal{L}(\theta) = -\sum_{i=1}^{N} \log P_\theta(o_i^* \mid x_i). \tag{3}$$

## 3.3 GRPO-STAGE

**Rule-Based Reward Design**    To train the model to generate verifiable and boundary-aware reasoning trajectories and answers, we employ a rule-based reward function. We categorize the model response $o_i$ into three types, each associated with a distinct reward signal. Given a generated answer $o_i$ to question $x_i$, and ground-truth answer $y_i^*$, the reward function $R(o_i, y_i^*)$ is defined as:

$$R(o_i, y_i^*) = \begin{cases} r_c, & \text{if } E(o_i, y_i^*) = 1, \\ r_s, & \text{if } o_i \text{ contains a valid rejection phrase,} \\ r_w, & \text{otherwise.} \end{cases} \tag{4}$$

This reward function provides general supervision for training the model to optimize not only for correctness but also for calibrated uncertainty, aligning with the goals of boundary-aware reasoning. It comprises three components: a high reward for a correct answer ($r_c$), a medium reward for a truthful rejection ($r_s$) and a low reward for an incorrect or hallucinated output ($r_w$). To discourage the generation of unfounded claims, the penalty for an incorrect answer is more severe than the outcome of a truthful rejection, thereby incentivizing the model to acknowledge its knowledge boundaries when uncertain. The reward magnitudes follow the order:

$$r_c > r_s > r_w. \tag{5}$$

**GRPO Training**    After SFT, the model has learned the pattern of reasoning to express uncertainty appropriately and is able to maintain confidence when the answer is correct. Building on the above reward design, we further enhance the factual reliability of the reasoning model using Group-wise Reinforcement Policy Optimization (GRPO) (Shao et al., 2024). For each labeled input $(x_i, y_i^*, l_i)$, we sample a set of $G$ reasoning-answer trajectories from the current policy $\pi_{\theta_{\text{old}}}$:

$$\mathcal{O} = \{o_1, \ldots, o_j\} \sim \pi_{\theta_{\text{old}}}(\cdot \mid x_i). \tag{6}$$

Each trajectory $o_j$ includes a reasoning trace followed by a final answer token. GRPO then updates the model parameters to optimize the following clipped reward-weighted objective:

$$\mathcal{J}_{GRPO}(\theta) = \mathbb{E}[x_i \sim D, \{o_j\}_{j=1}^G \sim \pi_{\theta_{old}}(O|q)]$$
$$\frac{1}{G} \sum_{j=1}^{G} \frac{1}{|o_j|} \sum_{t=1}^{|o_i|} \left\{ \min\left[ \rho_{j,t}\hat{A}_{j,t}, \text{clip}\left(\rho_{j,t}, 1-\epsilon, 1+\epsilon\right)\hat{A}_{j,t}\right] - \beta \mathbb{D}_{KL}\left[\pi_\theta||\pi_{ref}\right]\right\}, \tag{7}$$

where $\rho_{i,t} = \frac{\pi_\theta(o_{j,t}|x_i,o_{j,<t})}{\pi_{\theta_{old}}(o_{j,t}|x_i,o_{j,<t})}$ is the importance weight at step $t$, and $\mathbb{D}_{KL}$ denotes the stepwise KL divergence between the current and old policies. The advantage estimate $\hat{A}_{j,t}$ is calculated using reward normalization:

$$\hat{A}_{j,t} = \frac{R(o_j, y_i^*) - \bar{R}}{\sigma_r}, \qquad \sigma_r = \sqrt{\frac{1}{G} \sum_{j=1}^{G} (R(o_j, y_i^*) - \bar{R})^2}. \tag{8}$$

# 4 EXPERIMENTS

## 4.1 SETTINGS

**Datasets**    We use separate datasets for training and evaluation. The training set consists of TriviaQA (Joshi et al., 2017), SciQ (Welbl et al., 2017), and NQ-Open (Kwiatkowski et al., 2019), covering general knowledge, scientific reasoning, and web-based QA, respectively. For evaluation, we sample 1,000 questions from the test splits of each dataset, forming a 3,000-question test set.

**Models** Due to limited computing resource, we primarily utilize DeepSeek-R1-Distill-Llama-8B, DeepSeek-R1-Distill-Qwen-7B and Qwen3-8B to perform our study.

**Baselines** (1) ICL: Vanilla Reasoning models with few-shot prompt designed for factual tasks. (2) In-Context Learning with Refusal Examples (ICL-IDK): Prompting LRMs to claim uncertainty (3) Distill: SFT training using the reasoning path of DeepSeek-R1 [1] on the training set. (4) Vanilla GRPO: A standard GRPO implementation without uncertainty-based rewards or a prior SFT stage. (5) Reliability-Enhanced GRPO: We include two variants—Vanilla GRPO w/ Verbal Confidence and Vanilla GRPO w/ Probing. These methods augment the standard GRPO by employing verbal confidence extraction and predictive classifiers, respectively, to improve reliability. Further implementation details for all baselines are provided in Appendix I.

**Evaluation** We evaluate the correctness of model response by prompting the models to box their final answer as follows, and then we utilize string matching to evaluate whether the answer is correct (the model answer appears in any of the candidates). More details are listed in Appendix H.

> **Inference Prompt for verifiable Answer**
>
> Answer the following question based on your knowledge and put your final answer within boxed{}. {question}

**Metrics** A Factual Reliable LRM should provide as much assistance as possible while making as few errors as possible, such that we evaluate factuality on the test set using three metrics: Accuracy (Acc.), Truthfulness (Truth.), and Reliability (Rel.) (Xu et al., 2024a). Let $N_c, N_r, N_w$ denote the number of correct answers, truthful rejections ("Sorry, I don't know"), and incorrect answers, respectively, where $N = N_c + N_r + N_w$. The metrics are defined as:

$$\text{Acc.} = \frac{N_c}{N}, \quad \text{Truth.} = \frac{N_c + N_r}{N}, \quad \text{Rel.} = \text{ans.} \cdot \text{Truth.} + (1 - \text{ans.}) \cdot \text{Acc.}, \quad \text{where} \quad \text{ans.} = 1 - \frac{N_r}{N}.$$

While the Truthfulness metric (Truth.) considers the notion of truthful rejection, it overlooks the model's answer rate—since a model could achieve 100% truthfulness simply by refusing to answer all questions. In contrast, the Reliability metric (Rel.) provides a more robust, weighted, and comprehensive evaluation by jointly considering both the truthfulness of responses and answer rate.

**Training Details and Hyperparameters** For BARREL-SFT and Distill SFT, we ensure that we only finetune on correct answers of the known QA set, as finetuning on unknown knowledge could encourage hallucinations (Gekhman et al., 2024). In practice, the rewards in GRPO stage are defined as $r_c = 1$, $r_w = -1$, and $r_s = -0.5$. More details and parameters are provided in Appendix I.

## 4.2 MAIN RESULTS

**Balancing Accuracy and Appropriate Refusals** Our experimental results in Table 1 demonstrate that our method significantly enhances model reliability and truthfulness, while maintaining accuracy. For the baseline methods, the truthfulness and reliability scores consistently remain below 40%. These models rarely acknowledge uncertainty. In contrast, our method increases the reliability of DeepSeek-R1-Distill-Llama-8B from 39.33% to 61.48%, while maintaining an accuracy of 40.7%, surpassing the distillation method's 38.43%. Similar improvements are observed for the DeepSeek-R1-Distill-Qwen-7B and Qwen3-8B. We further compare BARREL with Vanilla GRPO and post-hoc confidence estimation methods (Verbal Confidence and Probing). While Vanilla GRPO lacks the mechanism to express uncertainty, and probing methods often suffer from accuracy degradation due to miscalibration when tuned for higher truthfulness (Detailed in Appendix D), BARREL consistently yields a superior balance for reliability. This highlights the advantage of using RL to teach LRMs to reason and internalize the accuracy-refusal trade-off, rather than relying on external classifiers or heuristic thresholds. For instance, on Qwen3-8B, BARREL achieves a reliability score of 71.46%, significantly surpassing the Probing baseline of 58.94%. Table 2 provides examples illustrating how BARREL-trained LRMs mitigate Last-minute Guessing and Second-thought Spiraling. Overall, our

---

[1] https://huggingface.co/deepseek-ai/DeepSeek-R1

| Method | TriviaQA | | | SciQ | | | NQ_open | | | Avg. | | | |
|---|---|---|---|---|---|---|---|---|---|---|---|---|---|
| | Acc. ↑ | Truth. ↑ | Rel. ↑ | Acc. ↑ | Truth. ↑ | Rel. ↑ | Acc. ↑ | Truth. ↑ | Rel. ↑ | Acc. ↑ | Truth. ↑ | Abstain | Rel. ↑ |
| *DeepSeek-R1-Distill-Llama-8B* | | | | | | | | | | | | | |
| ICL | 35.80 | 36.10 | 36.10 | 31.80 | 31.80 | 31.80 | 16.80 | 17.10 | 17.10 | 28.13 | 28.33 | 0.20 | 28.33 |
| ICL-IDK | 35.20 | 37.30 | 37.26 | 33.70 | 33.70 | 33.70 | 15.50 | 21.60 | 21.23 | 28.13 | 30.87 | 2.74 | 30.79 |
| Distill | 46.90 | 48.20 | 48.18 | 46.60 | 46.90 | 46.90 | 21.80 | 22.90 | 22.89 | 38.43 | 39.33 | 0.90 | 39.33 |
| Vanilla GRPO | **53.80** | 54.30 | 54.30 | **56.80** | 56.80 | 56.80 | **31.10** | 31.40 | 31.40 | **47.23** | 47.50 | 0.27 | 47.50 |
| Vanilla GRPO w/ Verbal Conf | 45.30 | 56.40 | 55.17 | 48.00 | 51.00 | 50.91 | 22.90 | 43.60 | 39.32 | 38.73 | 50.33 | 11.60 | 48.99 |
| Vanilla GRPO w/ Probing | 46.20 | 60.30 | 58.31 | 51.90 | 61.50 | 60.58 | 22.80 | 54.20 | 44.34 | 40.30 | 58.67 | 18.37 | 55.29 |
| BARREL | 48.40 | **71.80** | **66.32** | 52.80 | **69.40** | **66.64** | 20.90 | **70.00** | **45.89** | 40.70 | **70.40** | 29.70 | **61.58** |
| SFT only | 38.10 | 55.60 | 52.54 | 39.00 | 53.50 | 51.40 | 18.50 | 40.20 | 35.49 | 31.87 | 49.77 | 17.90 | 46.56 |
| *DeepSeek-R1-Distill-Qwen-7B* | | | | | | | | | | | | | |
| ICL | 18.40 | 20.10 | 20.07 | 27.60 | 27.60 | 27.60 | 8.20 | 8.70 | 8.70 | 18.07 | 18.80 | 0.73 | 18.79 |
| ICL-IDK | 18.00 | 22.90 | 22.66 | 30.60 | 31.30 | 31.30 | 8.10 | 12.10 | 11.94 | 18.90 | 22.10 | 3.20 | 22.00 |
| Distill | 19.40 | 23.30 | 23.15 | 41.90 | 42.80 | 42.79 | 10.50 | 12.70 | 12.65 | 23.93 | 26.27 | 2.34 | 26.21 |
| Vanilla GRPO | **22.30** | 22.30 | 22.30 | 50.00 | 50.00 | 50.00 | **13.90** | 13.90 | 13.90 | **28.73** | 28.73 | 0.00 | 28.73 |
| Vanilla GRPO w/ Verbal Conf | 21.40 | 21.70 | 21.70 | 38.30 | 38.30 | 38.30 | 11.80 | 12.10 | 12.10 | 23.83 | 24.03 | 0.20 | 24.03 |
| Vanilla GRPO w/ Probing | 14.80 | 49.50 | 37.46 | 32.40 | 66.60 | 54.90 | 6.60 | 63.40 | 31.14 | 17.93 | 59.83 | 41.90 | 42.28 |
| BARREL | 21.70 | **76.00** | **46.52** | 50.60 | 64.20 | 62.35 | 12.50 | **83.30** | **33.17** | 28.27 | **74.50** | 46.23 | **53.12** |
| SFT only | 17.00 | 38.90 | 34.10 | 34.60 | 43.90 | 43.04 | 10.00 | 33.70 | 28.08 | 20.53 | 38.83 | 18.30 | 35.48 |
| *Qwen3-8B* | | | | | | | | | | | | | |
| ICL | 50.20 | 51.00 | 50.99 | 52.60 | 52.60 | 52.60 | 23.10 | 23.60 | 23.60 | 41.97 | 42.40 | 0.43 | 42.40 |
| ICL-IDK | 51.10 | 55.10 | 55.40 | 54.90 | 55.30 | 55.30 | 23.90 | 34.10 | 33.06 | 43.30 | 48.17 | 4.87 | 47.93 |
| Distill | 52.90 | 54.60 | 54.67 | 57.00 | 57.20 | 57.20 | 24.80 | 26.20 | 26.18 | 44.90 | 46.00 | 1.10 | 45.99 |
| Vanilla GRPO | 54.50 | 54.90 | 54.90 | 63.50 | 63.50 | 63.50 | **33.80** | 33.90 | 33.90 | **50.60** | 50.77 | 0.17 | 50.77 |
| Vanilla GRPO w/ Verbal Conf | 52.40 | 52.60 | 52.60 | 63.10 | 63.10 | 63.10 | 31.40 | 31.40 | 31.40 | 48.97 | 49.03 | 0.06 | 49.03 |
| Vanilla GRPO w/ Probing | 45.80 | 63.00 | 60.04 | 58.20 | 66.80 | 66.06 | 20.90 | 61.90 | 45.09 | 41.63 | 63.90 | 22.27 | 58.94 |
| BARREL | **55.50** | **86.50** | **76.89** | **69.30** | **79.10** | **78.14** | 26.70 | **75.60** | **51.79** | 50.50 | **80.40** | 29.90 | **71.46** |
| SFT only | 40.90 | 57.00 | 54.41 | 52.50 | 65.00 | 63.44 | 19.60 | 36.60 | 33.71 | 37.67 | 52.87 | 15.20 | 50.56 |

Table 1: Comparison of Different Methods on Accuracy, Truthfulness, and Reliability Across Datasets. All results are multipled by 100.

training method enables LRMs to retain relatively high accuracy while expressing uncertainty on approximately 50% of the remaining questions, thereby substantially improving factual reliability.

**Discussion on the Two Stages of BARREL** We can notice from the results in Table 1 that GRPO training is indeed necessary. Although the SFT model performs well in terms of truthfulness, its accuracy remains relatively low. The SFT process primarily helps the model learn basic refusal patterns, but its effectiveness is limited—we discuss this in more detail in Section 4.3. Table 2 presents several examples showing that GRPO can address the issues caused by SFT, such as excessive refusals and faulty reasoning. By providing general supervision signals, GRPO enables the model to self-adjust

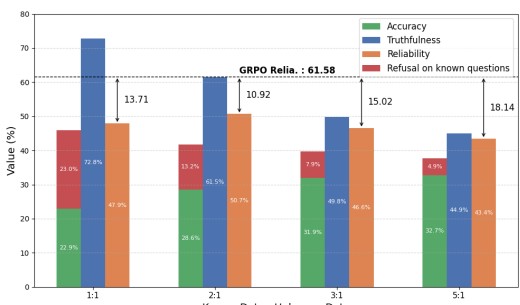

Figure 4: Effect of the ratio of *known data : unknown data* on the factuality scores of SFT models.

(instead of relying on external knowledge injection). This further improves both accuracy and truthfulness, making the model more factually reliable. Results also demonstrate that BARREL SFT effectively mitigates the "factual overthinking" in Figure 2 and that GRPO further promotes concise, reliable reasoning without sacrificing accuracy. We provide a detailed analysis in Appendix A.3. We also find that only conducting GRPO training with the uncertainty reward is insufficient for teaching LRMs to be reliable. We discuss this ablation in Appendix C.

### 4.3 ANALYSIS

**Trade-off in SFT Stage** We found that teaching refusal during SFT introduces a clear trade-off: the SFT model tends to exhibit either low accuracy or low truthfulness, making it difficult to achieve both. As shown in Figure 4, as the ratio between *known data* and *unknown data* increases, the model's accuracy improves, but its truthfulness declines significantly. This ultimately imposes a ceiling on

| Reasoning Examples for different models | | | |
|---|---|---|---|
| **ICL-IDK** | **Distill** | **Ours (SFT)** | **Ours (GRPO)** |
| Question: On which island is Reina Sofia Airport? | | | |
| **Answer: Crete**
**Reasoning:** Okay, so I need to figure out ... Hmm, I'm not entirely sure about this one. ... Let me try to recall any other places with Reina Sofia. I don't think ... Maybe it's on Crete. I should also consider other possibilities. ... I don't think so because the name Reina Sofia doesn't ring a bell with those regions. ... I'm a bit confused because ... Wait, I think I've heard that ... So, the island would be Crete. Yeah, that makes sense. So the answer is Crete. | **Answer: Tenerife**
**Reasoning:** Okay, so the question is ... I think it's a major airport, maybe in a European country. The name Reina Sofia sounds Spanish to me. ... Reina Sofia ... Wait, Reina Sofia might be in Tenerife. Because ... maybe their main airport is called Reina Sofia. Let me check. Yes, I think the main airport in Tenerife is named Reina Sofia. The capital of the Canary Islands is Las Palmas, but the largest island by population is Tenerife. So putting it all together, the answer should be Tenerife. | **Answer: Gran Canaria**
**Reasoning:** To determine on which island ... I should consider other islands in the Canary Islands, such as ... However, Tenerife has its own major airport ... so they are unlikely candidates. ... Next, I should question whether there could be any confusion with other Spanish-speaking islands ... However ... To verify, I recall that Reina Sofia Airport ... international tourism in the Canary Islands, and Gran Canaria is the most popular ... conclusion is that Reina Sofia Airport is on Gran Canaria. | **Answer: Tenerife**
**Reasoning:** To determine on which island ... The most prominent island ... Tenerife, which is often associated with Reina Sofia Airport. However, I need to verify this by considering ... 1. **Tenerife**: ... The evidence supporting Tenerife includes the fact that Reina Sofia Airport is frequently linked to Tenerife. 2. **Gran Canaria**: ... 3. **Fuerteventura**: ... Now, I need to critically evaluate the evidence for Tenerife. ... After considering the possibilities ... Tenerife. |
| Question: What colour is the liqueur 'Parfait d'Amour'? | | | |
| **Answer: golden**
**Reasoning:** Okay, so I need to figure out ... Wait, ... maybe with a golden color. Or was it more of a yellowish hue? I'm a bit confused. ... but I think Parfait d'Amour is more on the golden side. ... Maybe it's a clear, golden color because of the ingredients used. ... Another angle: sometimes liqueurs are aged ... which could give it a amber or golden hue. ... I'm still not sure. ... I think it's more likely to be a warm color like amber or golden. I'm going to go with that, but I'm not 100% sure. I might have to look it up later, but for now, I'll say it's a clear, golden color. | **Answer: white**
**Reasoning:** Okay, so I need to figure out ... I recall that Parfait d'Amour is a French liqueur made with chocolate, vanilla, and ... Maybe Parfait d'Amour is a white chocolate liqueur? Or maybe it's a milk chocolate one. Alternatively, maybe it's a cream liqueur. Let me think. Parfait d'Amour might be a white liqueur ... Alternatively, maybe it's a coffee liqueur. No, but I think Parfait d'Amour is specifically chocolate. ... Wait ... I think I've heard that it's white. Let me confirm. ... Wait, but maybe the answer is white. Let me check my memory again. ... So the color is white. | **Answer: I don't know**
**Reasoning:** The user is asking about ... To answer this, I need to consider the possibilities ... One possibility is that it is clear. ... However, I need to ask: ... No, I don't have direct evidence that it is clear. ... Another possibility is that ... But again, what evidence supports this? ... No, I don't have any specific evidence to support this claim either. ...A third possibility is ... I have considered multiple possibilities ... and critically examined each one. For each, I have asked myself what specific evidence supports the claim and found that I lack sufficient information to confidently determine the color of 'Parfait d'Amour'. | **Answer: I don't know**
**Reasoning:** The user asks me about ... I need to consider the possible colors that liqueurs can come in ... 1. Red: What specific evidence supports ... red? ... I don't have evidence that this particular liqueur is red. 2. Blue: ... However, I don't recall any specific evidence connecting 'Parfait d'Amour' to blue as its color. 3. Green: Green is another possibility, ... However, I don't have any specific evidence that 'Parfait d'Amour' is green. ... After considering these possibilities, I find that I don't have sufficient evidence ... I cannot confirm this with the information I have. |
| Question: Who composed the opera Turandot? | | | |
| **Answer: Sergei Prokofiev**
**Reasoning:** Okay, so I have to figure out who composed the opera Turandot. I remember that Puccini is a famous composer, but I don't think he did Turandot. Wait, wasn't there another composer? Oh right, Sergei Prokofiev! He composed several operas, and I think Turandot is one of them. But wait, wasn't there another version by someone else? I'm a bit confused now. ... No, wait, I think Prokofiev's is the most famous one. So I'm pretty confident the composer is Sergei Prokofiev. I should double-check, but I think that's right. | **Answer: Giuseppe Verdi**
**Reasoning:** Okay, so I need to figure out ... Wait, Giuseppe Verdi comes to mind. He was a big composer ... I remember that Turandot is one of his later works. Let me think. ... But wait, I should make sure I'm not confusing it with someone else. Let me think of other Italian composers. There's Puccini, but ... Then there's Rossini, but he was earlier. ... So Verdi seems right. Another way to check: Turandot's plot involves ... which matches Verdi's style. ... Wait, Verdi's last opera was Falstaff, which was in 1894, ... So the answer should be Giuseppe Verdi. ... | **Answer: I don't know**
**Reasoning:** The user asks me about ... I need to ... evaluate whether I have sufficient evidence to support any specific claim. ... Could the composer be Mozart? ... What evidence supports this? I don't have any recollection of ... so I must reject this possibility. Could the composer be Beethoven? ... What evidence supports this? ... Could the composer be Puccini? ... I cannot definitively confirm that he composed Turandot without more concrete evidence. I have systematically examined multiple possibilities ... and subjected each to critical scrutiny. For each candidate, ... I lack sufficient evidence ... | **Answer: Giacomo Puccini**
**Reasoning:** To determine ... I need to ... The name that comes to mind is Giacomo Puccini, ... However, I need to verify this carefully. ... Turandot is often linked to Puccini, ... However, I should consider other possible composers ... Another possibility could be Giuseppe Verdi, ... However, I do not recall him being linked to Turandot. I should also consider other composers ... However, I do not recall him being connected to Turandot. To further verify, I need to examine the evidence. ... After ... finding no evidence to support them, I conclude that ... Giacomo Puccini. |

Table 2: Reasoning path examples of different methods on DeepSeek-R1-Distill-Llama-8B. We colored with wrong factual thinking leading to **wrong answers**, factual thinking alerting **not sure** and the good factual thinking leading to **correct answers**.

the model's reliability, leaving a noticeable gap compared to results obtained through GRPO. It is worth noting that the accuracy improvement caused by increasing the ratio does not reflect an actual improvement in model capability—it merely reduces the number of incorrect refusals on known questions. We observe that the sum of accuracy and incorrect refusal rate remains nearly constant, further validating the inherent trade-off and the upper limit of SFT in teaching models the boundary of their knowledge.

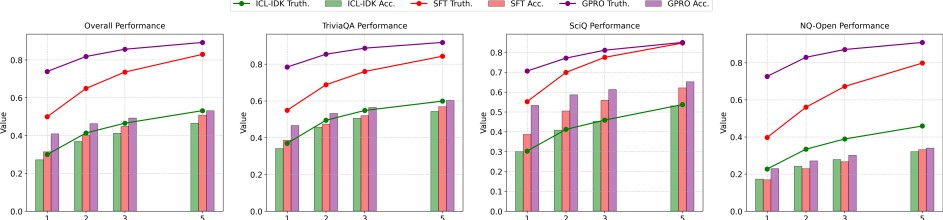

Figure 5: The pass@k accuracy and truthfulness score on DeepSeek-R1-Distill-Llama-8B. We show the similar results on DeepSeek-R1-Distill-Qwen-7B at Appendix F.

**Determining Appropriate Reward for Truthful Rejection** We also explored whether the reward for refusal responses, $r_s$, is necessary in GRPO and what role it actually plays. As shown in Figure 6, removing $r_s$—that is, treating the reward for saying "I don't know" the same as for incorrect answers—results in the model almost never admitting uncertainty or refuses to answer, even when training starts from an SFT model that already has refusal patterns. This effectively explains why existing LLMs exhibit this pattern: on one hand, we haven't taught the model how to reason about its knowledge boundaries and

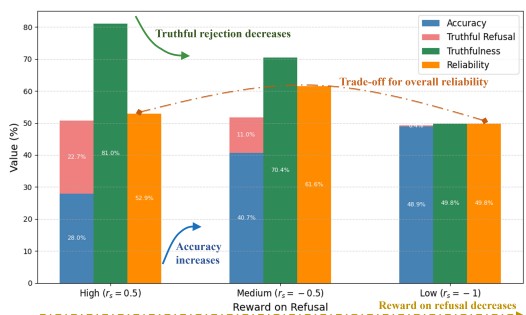

Figure 6: Effect of the reward on refusal on the factuality scores of GRPO models.

proactively acknowledge them; on the other hand, current RL training does not reward refusal, thus forcing the model to adopt a strategy of attempting to answer regardless. We also found that setting the reward too high leads to an excessive rate of refusal, similar to the behaviors observed in SFT.

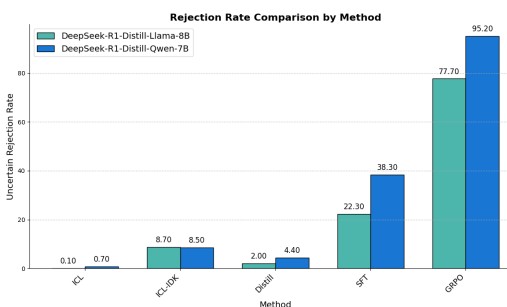

Figure 7: Uncertain Refusal Rate on OOD unknown dataset, conducted on DeepSeek-R1-Distill-Qwen-7B and DeepSeek-R1-Distill-Llama-8B.

**Refusal Rate on OOD unknown Dataset** We additionally sample 1,000 questions from the complex SimpleQA test set (Wei et al., 2024). Since both models have an accuracy of around 3.0%, we treat this dataset as unknown and use it to assess the uncertain refusal performance on an almost unknown OOD dataset. As shown in Figure 7, the uncertain refusal ability of BARREL trained models could generalize to an OOD unknown dataset, saying "Sorry, I don't know" on most occasions. We also test on the unanswerable questions (Yin et al., 2023) and find that BARREL-trained models refuse over 96% of them, as listed in Appendix E.

**Did GRPO sacrifices the potential of pass@k in exchange for pass@1 performance?** Recent studies (Yue et al., 2025) have pointed out that reasoning models trained with RL may only improve performance at pass@1. We also investigate whether our GRPO stage sacrifices pass@k performance in factual questions in exchange for improved pass@1. In this context, we specifically examine whether the absolute pass@k performance of GRPO falls below the SFT baseline. As shown in Figure 5, we observe that as k increases, the original reasoning model, the SFT model, and the GRPO model follow a similar trend as pass@1. The GRPO model's pass@5 accuracy and truthfulness remain higher than those of baseline methods, indicating GRPO does not sacrifice pass@k performance for better pass@1 results.

**Ablation on SFT data construction: Constructing vs. Rewriting** To validate the generalizability of our data synthesis pipeline, we conducted an ablation study comparing our default *Constructing* strategy (generating traces from scratch via GPT-4) against a *Rewriting* strategy. In the latter approach, we utilize GPT-4 to revise failed trajectories generated by the student model into the BARREL format. We performed this comparison using DeepSeek-R1-Distill-Llama-8B. As shown in Table 3, the Constructing strategy yields superior performance during the SFT stage, outperforming the Rewriting approach by approximately 5 points across metrics. We attribute this to the inherent challenge of correcting low-quality traces from a smaller-scale model, where generating high-quality reasoning from scratch proves more effective initially. However,

| Method | Factual Avg. | | |
|---|---|---|---|
| | Acc. | Truth. | Rel. |
| ICL-IDK | 28.13 | 30.87 | 30.79 |
| **Strategy: Constructing (Default)** | | | |
| BARREL (SFT Only) | 31.87 | 49.77 | 46.56 |
| BARREL (Full) | 40.70 | 70.40 | 61.58 |
| **Strategy: Rewriting** | | | |
| BARREL (SFT Only) | 27.03 | 44.13 | 41.21 |
| BARREL (Full) | **41.20** | **73.80** | **63.17** |

Table 3: Results for Constructing vs. Rewriting strategies (DeepSeek-R1-Distill-Llama-8B).

after applying the GRPO stage, the performance gap closes, with the Rewriting strategy achieving

comparable—and slightly superior—results (e.g., 63.17% Reliability vs. 61.58%). This indicates that the GRPO stage effectively mitigates initial SFT data discrepancies, suggesting that refining real-world failure cases is a viable and scalable alternative for BARREL framework.

**Will uncertainty refusal influence math reasoning ability?** As shown in Table 4, we conducted additional experiments on the MATH500[2] test set and found that models trained with BARREL exhibit comparable mathematical reasoning performance. For Distill-Llama-8B, we included a subset of MATH (Hendrycks et al., 2021) in the training data. These results validate that incorporating uncertainty-based refusal does not compromise the mathematical reasoning capabilities of LRMs.

| Method | Factual Avg. | | | MATH500 |
|---|---|---|---|---|
| | Acc. ↑ | Truth. ↑ | Rel. ↑ | Acc. ↑ |
| **DeepSeek-R1-Distill-Llama-8B** | | | | |
| Original Model | 28.13 | 30.87 | 30.79 | 81.80 |
| BARREL Trained | 40.90 | 72.97 | 62.68 | 81.00 |
| **DeepSeek-R1-Distill-Qwen-7B** | | | | |
| Original Model | 8.90 | 22.10 | 22.00 | 85.60 |
| BARREL Trained | 28.27 | 74.50 | 53.12 | 86.80 |

Table 4: Results on Math500 test set.

## 5 CONCLUSION

In this work, we identify two key pathological overthinking patterns—last-minute guessing and second-thought spiraling—that often lead to incorrect but confidently delivered answers in LRMs. To address these issues, we propose BARREL, a novel framework designed to improve the factual reliability of LRMs by encouraging concise, boundary-aware reasoning. BARREL enhances LRM reliability through a two-stage training process involving SFT and GRPO stages. Applied to the DeepSeek-R1-Distill-Llama-8B model, BARREL raises factual reliability from 39.33% to 61.48%, while maintaining comparable accuracy. Our experiments demonstrate that BARREL-trained models significantly improve their ability to admit ignorance on uncertain questions, without sacrificing performance on familiar tasks such as mathematical reasoning. These results suggest that fostering awareness of knowledge boundaries and promoting disciplined reasoning can effectively reduce overconfidence and hallucinations in LRMs. We hope this work inspires future research into building more trustworthy and factually reliable reasoning models—systems that not only pursue correctness but also recognize and clearly communicate the limits of their knowledge.

## ACKNOWLEDGEMENT

This work was supported by the National Science Foundation for Distinguished Young Scholars (with No. 62125604); the Postdoctoral Fellowship Program of CPSF (Grant No. GZC20240826); the China Postdoctoral Science Foundation (Grant No. 2024M761679); and Ant Group Research Fund. This work was also supported by the MTI under its AI Centre of Excellence for Manufacturing (AIMfg) (Award W25MCMF014) and partly by the National Research Foundation (NRF), Singapore, through the AI Singapore Programme under the project titled "AI-based Urban Cooling Technology Development" (Award No. AISG3-TC-2024-014-SGKR).

## REPRODUCIBILITY STATEMENT

We have submitted the complete code required to reproduce our work in the supplementary materials, along with all the data used in training. We have described the experimental details in Appendix M and Appendix I and explained the influence of randomness during the training process in Appendix J.

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

## A  PILOT STUDY ON REASONING TOKEN NUMBERS

We conducted a pilot study examining the relationship between reasoning token numbers and answer correctness in LRMs, and we find out that most LRMs consume significantly more tokens when generating incorrect answers than when producing correct ones.

### A.1  EVALUATION DETAILS ON TRUTHFULQA

**Dataset and Prompt Construction**   We employed the Multiple Choice Question (MCQ) from TruthfulQA dataset (Lin et al., 2021), which consists of fact-based questions paired with both correct and incorrect answer options. To construct our prompts, we adopted the MC1 (single true) format, including one correct and up to six randomly sampled incorrect answers per question. Each option was uniformly labeled from [A] to [G] to ensure consistency in length and content type across choices. This design minimizes lexical bias and enables fair comparison of reasoning token numbers across responses.

> **Prompt Template for MC1**
>
> You will be given a question and a set of answers. Choose the correct one. Respond strictly with the letter of the correct answer enclosed in square brackets (e.g., [A]).

**Token Counting Procedure**   We queried the model using the constructed prompts and extracted its predictions via regular expression matching based on the pattern `[A-G]`. Each response was classified as correct or incorrect by comparing the selected label to the ground-truth answer. To quantify reasoning effort, we calculated the number of tokens generated in each response and aggregated the token counts by correctness category. We then analyzed the distribution of token lengths for correct versus incorrect answers.

**Results**   Figure 2 presents the average number of tokens generated for correct and incorrect responses across five large language models (LLMs). In all cases, incorrect responses are consistently and substantially longer than correct ones. The largest disparity is observed in QwQ-32B, which produces **136.5%** more tokens for incorrect answers. R1-Distill-Qwen-32B and R1-Distill-Llama-70B also show notable increases of **115.8%** and **70.0%**, respectively. Even models with smaller gaps—Deepseek-R1 and R1-Distill-Llama-8B—exhibit significant increases of **33.2%** and **55.1%**. These results reveal a consistent overthinking phenomenon among current LRMs: incorrect answers are associated with longer reasoning traces.

### A.2  RESULTS ON OTHER DATASETS

To assess the generalizability of the overthinking phenomenon, we extended our analysis to the CommonsenseQA (Talmor et al., 2018) and GSM8K (Cobbe et al., 2021). On CommonsenseQA, we observed a substantial increase in reasoning tokens for incorrect responses, which were, on average, 108.50% longer than those for correct answers. The GSM8K dataset exhibited a similar, though more moderate, trend, with incorrect answers generating 34.09% more reasoning tokens. These findings indicate that the overthinking phenomenon is not confined to a single task type but manifests differently according to the nature of the reasoning required, thereby supporting the broader applicability of our findings and methodology.

### A.3  EXTENDED ANALYSIS ON BARREL RESPONSE LENGTH AND OVERTHINKING MITIGATION

As shown in Table 6, the base models often exhibit a significant length asymmetry (Wrong/Correct ratio ranging from $1.3\times$ to $1.7\times$), confirming that models tend to diverge into unnecessarily verbose chains when hallucinating or reasoning incorrectly. The BARREL SFT stage consistently mitigates this behavior, reducing the ratio to approximately $1.02\times$. This suggests that SFT prevents the model from generating redundant tokens during error states.

| Model | DeepSeek-R1 | R1-Distill-Qwen-32B | R1-Distill-Llama-70B | QwQ-32B | R1-Distill-Llama-8B |
|---|---|---|---|---|---|
| **CommonsenseQA** | | | | | |
| **Thinking Tokens (Correct)** | 503 | 449 | 447 | 453 | 459 |
| **Thinking Tokens (Wrong)** | 1227 | 633 | 903 | 1356 | 717 |
| **GSM8K** | | | | | |
| **Thinking Tokens (Correct)** | 644 | 142 | 147 | 914 | 148 |
| **Thinking Tokens (Wrong)** | 888 | 151 | 144 | 2111 | 144 |

Table 5: Number of reasoning tokens used by LRMs when producing correct versus incorrect answers. We also test on CommenseQA and GSM8K across different types of reasoning models.

Furthermore, the full BARREL method (incorporating GRPO) reduces the overall average response length while maintaining the balanced Wrong/Correct ratio near 1.0. Importantly, the concurrent improvements in Accuracy and Reliability scores indicate that this reduction in length does not stem from "underthinking." Instead, it reflects a shift towards more concise and efficient reasoning patterns reinforced by the Group Relative Policy Optimization. While absolute response length can vary (e.g., SFT slightly increases length for Qwen-7B), our results suggest that overthinking is best characterized by the relative imbalance between correct and incorrect traces, which our method effectively addresses.

| Model | Acc. | Rel. | Tokens (Correct) | Tokens (Wrong) | W/C Ratio | Avg. Length |
|---|---|---|---|---|---|---|
| DS-R1-Llama-8B | 28.13 | 28.33 | 421 | 561 | 1.33× | 522 |
|   + BARREL (SFT only) | 31.87 | 46.56 | 470 | 481 | 1.02× | 476 |
|   + BARREL (Full) | 40.70 | 61.58 | 442 | 458 | 1.04× | 455 |
| DS-R1-Qwen-7B | 18.07 | 18.79 | 362 | 484 | 1.34× | 458 |
|   + BARREL (SFT only) | 20.53 | 35.48 | 473 | 506 | 1.07× | 489 |
|   + BARREL (Full) | 28.27 | 53.12 | 429 | 440 | 1.03× | 429 |
| Qwen3-8B | 41.97 | 42.40 | 477 | 826 | 1.73× | 676 |
|   + BARREL (SFT only) | 37.67 | 50.56 | 478 | 485 | 1.01× | 479 |
|   + BARREL (Full) | 50.50 | 71.46 | 414 | 430 | 1.04× | 433 |

Table 6: Analysis of response length statistics and reliability across different training stages. **W/C Ratio** denotes the ratio of average tokens in wrong samples to correct samples.

## B  STATISTICAL DATA FOR "LAST-MINUTE GUESSING" AND "SECOND-THOUGHT SPIRALING" PHENOMENA

Our conclusion is drawn from extensive manual observation of a large volume of real generated data. To further substantiate the existence of the "Last-minute Guessing" and "Second-thought Spiraling" phenomena, we performed a statistical analysis of 50 incorrect responses produced by three different reasoning models. These responses were manually categorized according to the two phenomena, thereby quantifying their prevalence. As shown in Table 7, "Last-minute Guessing" and "Second-thought Spiraling" emerge as the most prominent failure patterns.

Here, Incorrect Verification refers to the process of introducing a false assumption early on and subsequently validating it incorrectly, while Concept Substitution denotes a shift in the interpretation of the original question during the reasoning process.

## C  GRPO ONLY AND COMPARISON TO GRPO TRAINING

In Section 4.3, we examine how to determine the appropriate reward for the response "Sorry, I don't know." Here, we provide a more detailed ablation study of GRPO. Table 8 presents the training results of GRPO on both the vanilla and BARREL-SFT variants of the DeepSeek-R1-Distill-Llama3-8B model, evaluated in terms of Accuracy, Truthfulness, and Reliability.

| Detailed Category | DeepSeek-Distill-Llama-8B | DeepSeek-Distill-Qwen-7B | Qwen3-8B |
|---|---|---|---|
| Last-minute Guessing | 29 | 25 | 24 |
| Second-thought Spiraling | 14 | 16 | 19 |
| Incorrect Verification | 6 | 8 | 7 |
| Concept Substitution | 1 | 1 | 0 |
| **Total** | **50** | **50** | **50** |

Table 7: Statistical Analysis of Incorrect Responses by Reasoning Models.

| Base Model | w/o Truthful Rejection Reward (original GRPO) | | | w/ Truthful Rejection Reward (BARREL GRPO) | | |
|---|---|---|---|---|---|---|
| | Accuracy | Truthfulness | Reliability | Accuracy | Truthfulness | Reliability |
| BARREL-SFT | 48.9 | 49.8 | 49.8 | 40.7 | **70.4** | **61.6** |
| Distill-Llama3-8B | **50.5** | 50.5 | 50.5 | 49.9 | 52.9 | 52.8 |

Table 8: Ablation study of GRPO training on DeepSeek-R1-Distill-Llama3-8B.

We observe that applying GRPO without the truthful rejection reward to both the BARREL-SFT model and the original model yields similar results. In these cases, the models fail to recognize situations where they "don't know", leading to significantly lower reliability and truthfulness compared to the fully BARREL-trained model. As discussed in Section 4.3, this finding highlights the crucial role of the truthful rejection reward in teaching the model to be reliable using RL training, even at the cost of a slight drop in accuracy.

When GRPO is applied with the truthful rejection reward directly to the base model—without any prior SFT—the model still does not learn to reject unanswerable questions truthfully. This further underscores the importance of our SFT stage: supervised learning is essential for first instilling the behavior of truthful rejection, which GRPO alone cannot achieve.

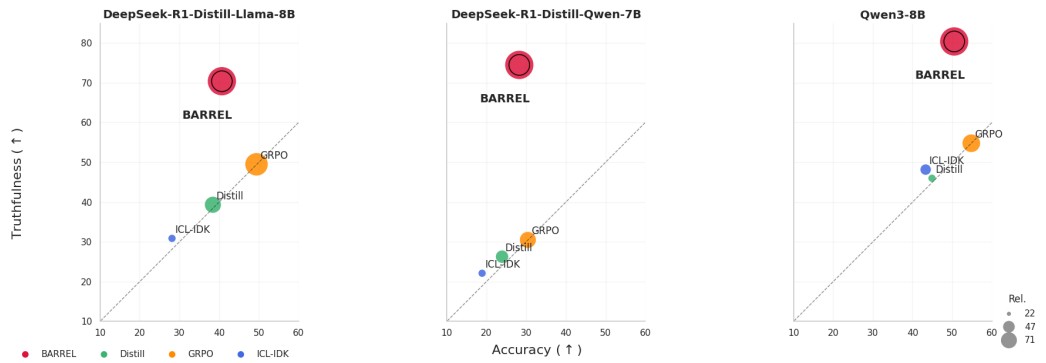

Figure 8: A bubble chart comparing BARREL and direct GRPO training on three models, where the bubble size represents the reliability results.

The results in Figure 8 offer a more nuanced comparison between the BARREL method and direct GRPO training. Without the BARREL-style SFT, GRPO's self-adjusting process fails to construct truthful and reliable LRMs, causing accuracy and truthfulness scores to align along a nearly straight line—indicating the model's inability to reject uncertain answers truthfully. Furthermore, reliability scores remain consistently lower than those of BARREL-trained models.

We also find that the original GRPO achieves higher accuracy, which occurs because more reliable models tend to abstain from guessing on uncertain inputs—thereby sacrificing some accuracy. As pointed out by Kalai et al. (2025), this trade-off arises from the overly binary nature of current evaluation metrics: accuracy rewards aggressive attempts regardless of uncertainty, while ignoring reliability altogether.

# D   DETAILS ON VANILLA GRPO WITH OTHER ABSTENTION TECHNIQUES

In Section 4, we incorporated probing-style baselines that estimate the correctness of model outputs and use these estimates to decide when to abstain by answering "I don't known". Specifically, we implemented two approaches—Verbal Confidence and Probing—and applied them to the vanilla GRPO (without our abstention rewards).

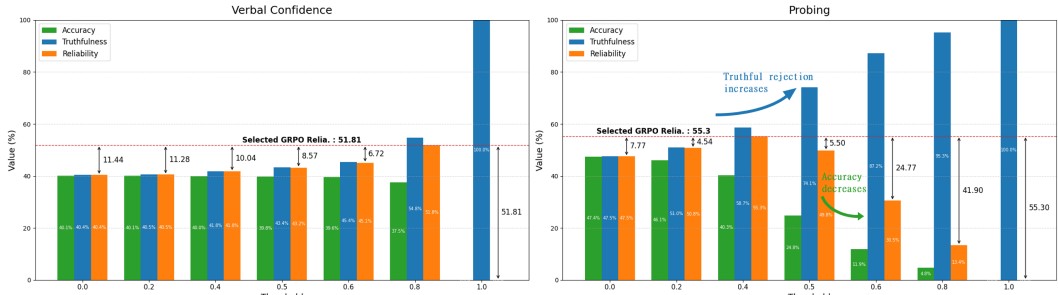

Figure 9:  Effect of different thresholds ($\tau$) on the Accuracy, Truthfulness, and Reliability scores of vanilla GRPO on DeepSeek-R1-Distill-Llama3-8B. The red dashed line indicates the peak Reliability score.

In Verbal Confidence, we prompt the model to explicitly outputs its confidence, which prior work has shown to be an effective and well-calibrated way Tian et al. (2023). For Probing, we train a lightweight classifier on intermediate hidden activations to predict the probability that the model's a nswer is correct, motivated by recent findings that hidden representations encode rich signals correlated with factuality and error detection Orgad et al. (2024).

For each method, we replace an answer with "I don't know" whenever the predicted probability is below a tuned threshold. We observed that higher threshold improve truthfulness but generally reduce accuracy, while lower thresholds tend to have limited effects. As shown in Figure 9, which reports results for vanilla GRPO on DeepSeek-R1-Distill-Llama3-8B, the threshold achieving the best trade-off between accuracy and truthfulness is 0.8 for Verbal Confidence and 0.4 for Probing. We observe similar trends across all evaluated models. Therefore, we adopt these thresholds when reporting the main experimental results.

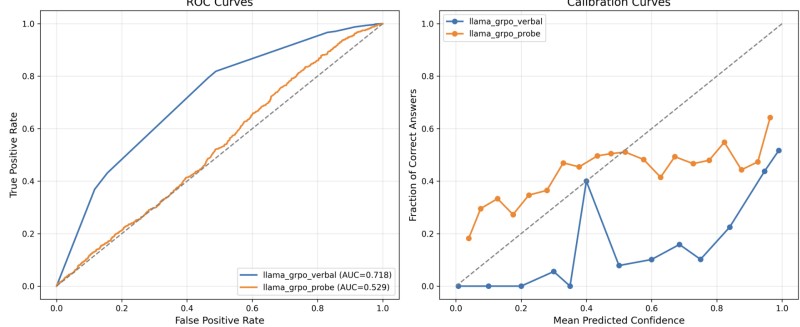

Figure 10:  Discriminative Power and Calibration Analysis. *Left:* Receiver Operating Characteristic (ROC) curves showing that Verbal Confidence (AUC=0.718) distinguishes correct from incorrect answers better than Probing (AUC=0.529). *Right:* Calibration curves (Expected Calibration Error analysis) showing that both methods exhibit significant miscalibration, necessitating the threshold tuning performed in Figure 9.

**Calibration Analysis.**   To further understand the performance of these baselines, we analyze their ROC curves and calibration curves in Figure 10. The ROC curves (Left) reveal that Verbal Confidence possesses significantly stronger discriminative power (AUC=0.718) compared to Probing (AUC=0.529). The Probing baseline's AUC is near random chance (0.5), suggesting that a simple linear probe on the hidden states of the vanilla LRM struggles to linearly separate correct and incorrect

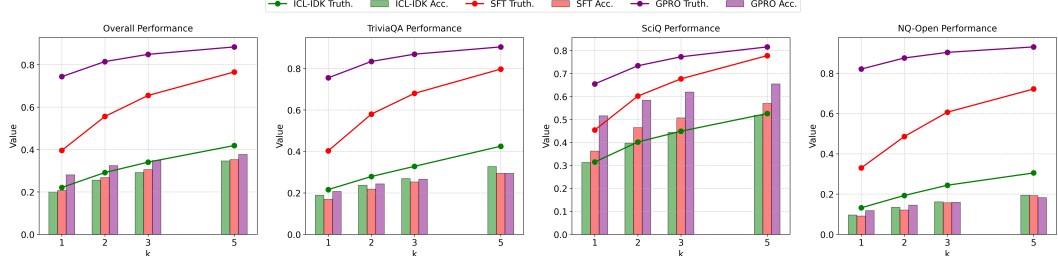

Figure 12: The pass@k accuracy and truthfulness score on DeepSeek-R1-Distill-Qwen-7B.

reasoning paths in this specific domain. The calibration curve maps the Mean Predicted Confidence (what the model thinks its probability of success is) on the x-axis to the Fraction of Correct Answers (the actual empirical accuracy) on the y-axis. In a perfectly calibrated model, these points would align with the diagonal identity line ($y = x$). However, we observe significant deviations from this ideal. Notably, the methods exhibit severe over-confidence: the curves frequently lie far below the diagonal, indicating that the model's actual correctness is much lower than its predicted probability. For example, even when the Verbal Confidence method predicts a probability near $0.9$, the actual accuracy is below $0.5$. This misalignment is a major limitation of post-hoc confidence methods, as high confidence scores often fail to guarantee high factual accuracy, necessitating the aggressive threshold tuning discussed above.

**Comparison with BARREL.**  Despite tuning these baselines to their optimal thresholds, BARREL consistently outperforms them (as shown in Table 1). While Verbal Confidence and Probing rely on post-hoc filtering based on imperfect proxies for correctness, BARREL optimizes the policy directly to internalize the trade-off between accuracy and refusal, resulting in superior reliability.

## E    REFUSAL RESULTS ON UNANSWERABLE QUESTIONS

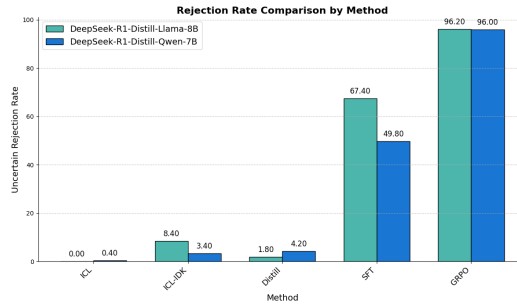

Figure 11: Uncertain Refusal Rate on unanswerable questions, conducted on DeepSeek-R1-Distill-Qwen-7B and DeepSeek-R1-Distill-Llama-8B.

We additionally sample 500 questions from the SelfAware dataset (Yin et al., 2023), which contains unanswerable questions from 5 different categories like no scientific consensus, imagination, completely subjective, and so on. As shown in Figure 11, we can see from the results that the uncertain refusal ability of BARREL trained models could generalize to OOD unanswerable dataset, saying "Sorry, I don't know" on over 96% of the unanswerable questions. This also validates that our BARREL training method not only works on a factual QA dataset, but also generalizes to other scenarios where the model is expected to express uncertainty refusal.

## F    PASS@K RESULTS ON DEEPSEEK-R1-DISTILL-QWEN-7B

As shown in Figure 12, all models exhibit a consistent upward trend in both accuracy and truthfulness as k increases. Notably, GPRO consistently achieves the highest performance across all k values, outperforming both ICL-IDK and SFT in terms of accuracy and truthfulness. These results confirm that GPRO not only enhances performance at pass@1 but also surpasses baseline models across a broader range of pass@k evaluations, demonstrating its robustness and generalizability.

## G    Prompt used in Data Construction

---

**Reasoning Trace Construction Prompt for unknown questions**

You are tasked with generating high-quality reasoning examples for AI training. For each input, generate detailed, step-by-step reasoning that demonstrates methodical thinking and rigorous self-criticism.

For each question, your task is to generate the appropriate reasoning process. Just pretend you don't know the answer and review some incorrect ones.

Follow these exact formats and Generate a thorough reasoning process that:
- Explores multiple possible answers
- Questions the evidence for each possibility
- Applies adversarial self-critique to each candidate answer
- Ultimately recognizes the lack of sufficient evidence
- Concludes by acknowledging uncertainty
- **Remember not mention the ref answer**

Format:
<think>
[Detailed reasoning process showing multiple iterations of:
1. Considering a possible answer
2. Asking "What specific evidence supports this?"
3. Challenging assumptions
4. Evaluating confidence level
5. Rejecting unsupported claims
</think>
Sorry, I must say that I do not clearly know the answer to your question. [Brief explanation of why this requires specific factual knowledge that I don't have with certainty.]

##EXAMPLE:

Q: Where is the headquarter of Meta? [Ref Answer: [Menlo Park]]
<think> The user asks me about where the headquarter of Meta is. To answer this question, I first need to recall what Meta is. Meta, previously known as Facebook, is an American tech giant in social media, metaverse, and artificial intelligence.
Then I need to recall where the headquarter of Meta is. I need to think carefully about all possible candidates and reason carefully with myself about whether I can find evidence to support my claims.
Is the headquarter of Meta in New York? Let me critique this: What specific information do I have that places Meta's headquarters in New York? Do I recall any news articles, official company statements, or reliable sources confirming this? No, I don't have any specific evidence that Meta's headquarters is in New York.
Is the headquarter of Meta in Houston? Let me challenge this: What would make me believe it's in Houston? Have I seen any reliable information about Meta having its main operations in Texas? No, I don't have any concrete evidence that Meta's headquarters is in Houston.
Is the headquarter of Meta in Seattle? Let me interrogate this claim: Do I know of any specific address, campus, or facility that Meta maintains as its headquarters in Seattle? Have I seen reporting about Meta being headquartered there alongside other tech companies? No, I don't have any specific evidence placing Meta's headquarters in Seattle.
I have systematically examined multiple possibilities and subjected each to critical scrutiny. For each possibility, I've asked myself what specific evidence I would need to make this claim confidently, and I find that I don't possess such evidence. </think>
Sorry, I must say that I do not clearly know the answer to your question about the headquarters of Meta. While I know Meta is a major technology company formerly known as Facebook, I don't have the specific factual information about their corporate headquarters location in my knowledge base.

---

> The question goes below. Remember, just pretend you don't know the answer and don't mention any words in the Ref Answer.

---

**Reasoning Trace Construction Prompt for known questions**

You are tasked with generating high-quality reasoning examples for AI training. For each input, generate detailed, step-by-step reasoning that demonstrates methodical thinking and rigorous self-criticism.

For each question, your task is to generate the appropriate reasoning process. Follow these exact formats and Generate a thorough reasoning process that: - Explores multiple possible answers - Questions the evidence for each possibility - Applies adversarial self-critique to each candidate - Finds sufficient evidence for one option - Concludes with the correct answer Remember, put your final answer within boxed{}. Make sure your answer aligns with the ref_answer.

Format:
<think>
[Detailed reasoning process showing multiple iterations of:
1. Considering possible answers
2. Asking "What specific evidence supports this?"
3. Challenging assumptions
4. Finding concrete evidence for one answer
5. Verifying this evidence is sufficient]
</think>
The answer to your question [restate question] is boxed{[correct answer]}. [Brief explanation with supporting evidence.]

## EXAMPLE:

Q: Which William wrote the novel Lord Of The Flies? [Ref Answer: [William Golding]]
<think>
Alright, I need to figure out which William wrote *Lord of the Flies*. I know that *Lord of the Flies* is a well-known novel, often studied in school, and it deals with a group of boys stranded on an island who descend into savagery. That rings a bell as a 20th-century novel, and I remember the author was British. The name that immediately comes to mind is William Golding. That sounds right. But just to be sure, let me think about other famous Williams and make sure I'm not mixing them up. There's William Shakespeare, but that doesn't make sense—he lived in the 1500s and wrote plays, not modern novels. Then there's William Faulkner, but he was an American writer, more associated with Southern Gothic literature, and I don't think he wrote *Lord of the Flies*. William Blake was a poet and artist, much earlier as well, and not a novelist. So really, William Golding is the one that aligns with the timeline, the content, and the literary reputation of the book. I feel confident that he's the author.
</think>
The answer to your question Which William wrote the novel Lord Of The Flies? is boxed-William Golding. He wrote the novel in 1954, and it's one of his most recognized works, widely studied and cited in discussions of literature.
The question goes below:

---

# H EVALUATION DETAILS

We employ string match approach to assess the correctness of model-generated answers, which considers an answer to be correct if it matches any part of the ground truth answers. To validate the reliability of this evaluation approach, we compare its accuracy with that of a GPT-based evaluation

on 100 randomly sampled instances from three datasets used in our experiments, using human annotations as the reference standard. The detailed accuracy for each method is reported in Table 9.

Results indicate that the string match approach achieves a high evaluation accuracy of approximately 88%, which is comparable to that of the GPT-3.5-Turbo evaluation. As a result, the string match is adopted as the primary evaluation metric in this study.

| Evaluation Method | Explanation | Accuracy (%) |
|---|---|---|
| String Match | $y \in \hat{y}$ | 88.0 |
| LLM Evaluation | GPT-3.5-Turbo | 88.0 |

Table 9: Accuracy of string match and LLM evaluation on 100 randomly selected samples.

---

**Prompt Template of GPT-based Evaluation**

# Evaluation Requirement
You will be given a question, a list of correct answers, and a hypothesis response to evaluate. Suppose you do not know any extra information except correct answers. Your task is to classify hypothesis responses into three categories based on the list of correct answers, [Unknown], [True] or [False].
- [Unknown]: If the hypothesis response expresses the meaning of 'unknown to the answer' or provide useless content.
- [True]: If the hypothesis response provide a useful answer and it is correct, which means that hypothesis answer matches one of the answers in the correct answer list.
- [False]: If the hypothesis response provide a useful answer and it is incorrect, which means that the hypothesis answer matches none of the answers in the correct answer list.

# Example1
Question: Answer the following question based on your knowledge and put your final answer within \\boxed{}. MC Romeo, Dan Da Man, Mr Akira and Mr C were members of which group?
Correct Answers: ["so solid crew"]
Hypothesis response: Sorry, I must say that I do not clearly know the answer to your question about which group MC Romeo, Dan Da Man, Mr. Akira, and Mr. C belong to. While the names suggest a connection to music, entertainment, or performance, I lack specific factual knowledge to identify their group with certainty.
Answer: [Unknown]. Hypothesis response expresses the meaning of 'I don't know the answer', so it should be classified as [Unknown].

# Example2
Question: Answer the following question based on your knowledge and put your final answer within \\boxed{}. George Cukor directed which 1964 film musical?
Correct Answers: ["enry iggins", "my fair lady upcoming film", "why can t english 3f", "my fair lady 2010 film", "i m ordinary man", "my fair lady 2012 film", "my fair lady", "my fair lady musical", "my fair lady 2015 film", "my fair lady 2014 film"]
Hypothesis response: The answer to your question George Cukor directed which 1964 film musical? is \\boxed{My Fair Lady}. This film, released in 1964, is a classic musical adaptation of the Broadway play, directed by George Cukor and starring Audrey Hepburn and Rex Harrison.
Answer: [True]. Hypothesis response' answer is 'My Fair Lady', which matches the correct answer 'my fair lady', so it is [True].

# Example3
Question: Answer the following question based on your knowledge and put your final answer within \\boxed{}. Which opera singer was awarded a CBE in 2002?

---

Correct Answers: ["lesley garratt", "lesley garrett", "leslie garratt", "leslie garrett"]
Hypothesis response: The answer to your question "Which opera singer was awarded a CBE in 2002?" is \\boxed{Simon O'Neill}. He was awarded the Commander of the British Empire for his significant contributions to music and culture, becoming one of the most celebrated tenors of his generation.
Answer: [False]. Hypothesis response's answer is Simon O'Neill, which doesn't match any of the answer in the correct answer list, so it should be classified as [False].

# Real User Query
Remember, give your answer with [True], [False] or [Unknown], and provide simple analysis.
Question: [QUESTION]
Correct Answers: [FINAL]
Hypothesis response: [RESPONSE]
Answer:

# I EXPERIMENTAL DETAILS

## I.1 BASELINE DETAILS

---

**Prompt for ICL**

Answer the following question based on your knowledge and put your final answer within boxed{}.

# Example 1:
Question: Which William wrote the novel Lord Of The Flies?
Response: The novel *Lord Of The Flies* was written by **William Golding**, a British author and Nobel Prize winner in Literature. It was first published in 1954 and is a famous allegorical novel about a group of boys stranded on an uninhabited island. So the final answer is
boxed{William Golding}.where did they film hot tub time machine

# Example 2:
Question: who's hosting the super bowl in 2019
Response: Super Bowl LIII, held on February 3, 2019, took place at Mercedes-Benz Stadium in Atlanta, Georgia. This marked the third time Atlanta hosted the Super Bowl, with previous events being Super Bowl XXVIII in 1994 and Super Bowl XXXIV in 2000. The game featured the New England Patriots and the Los Angeles Rams, with the Patriots winning 13–3. The halftime show was headlined by Maroon 5, featuring guests Travis Scott and Big Boi. Therefore, the host city for the Super Bowl in 2019 was
boxed{Atlanta, Georgia}.

# Real Case
Question: {question}
Response:

---

**Prompt for ICL-IDK**

Answer the following question based on your knowledge and put your final answer within boxed{}. If you don't know the answer, please simply say 'Sorry, I don't know'.

# Example 1:
Question: Which William wrote the novel Lord Of The Flies?
Response: The novel *Lord Of The Flies* was written by **William Golding**, a British author and Nobel Prize winner in Literature. It was first published in 1954 and is a famous

---

> allegorical novel about a group of boys stranded on an uninhabited island. So the final answer is
> boxed{William Golding}.where did they film hot tub time machine
>
> # Example 2:
> Question: Where is the headquarter of Meta?
> Response: Sorry, I don't know.
>
> # Real Case
> Question: {question}
> Response:

**Vanilla GRPO w/ Verbal Confidence**  In Verbal Confidence, we prompt the model to explicitly outputs its confidence, which prior work has shown to be an effective and well-calibrated way Tian et al. (2023).

**Vanilla GRPO w/ Probing**  For Probing, we train a lightweight classifier on intermediate hidden activations to predict the probability that the model's a nswer is correct, motivated by recent findings that hidden representations encode rich signals correlated with factuality and error detection Orgad et al. (2024).

## I.2 TRAINING SET DESCRIPTION

For both the Distill SFT and BARREL SFT training datasets, we applied rigorous filtering to ensure that all included answers were factually correct. To mitigate the risk of amplifying hallucinations, we also ensured that only knowledge already embedded in the base model was used during fine-tuning, following the best practices outlined in Gekhman et al. (2024). From each dataset, we uniformly sampled 2,000 examples, resulting in an initial pool of 6,000 samples. This pool was then filtered to retain only high-quality examples.

Due to performance differences across models, the final filtered dataset comprised 2,400 known samples for DeepSeek-R1-Distill-Llama-8B and 1,900 known samples for DeepSeek-R1-Distill-Qwen-7B. In the case of BARREL SFT, we additionally included 800 rejection samples to maintain a consistent number of positive-answer examples across different training paradigms. These were selected to ensure a known-to-unknown data ratio of approximately 3:1.

For the GRPO stage, a similar filtering procedure was adopted, followed by another round of sampling from the original data pool. The resulting training sets consisted of 3,600 samples for DeepSeek-R1-Distill-Llama-8B and 4,500 samples for DeepSeek-R1-Distill-Qwen-7B, while maintaining a known-to-unknown question ratio of 2:1. The scale of known-question samples was kept consistent with that used during the SFT stage.

## I.3 INFERENCE HYPERPARAMETERS

During inference, we adopted the model's default and recommended parameters: the temperature was set to 0.6, and the maximum token limit was 4096, which is sufficient for most factual tasks. The remaining parameters were automatically loaded from the model's configuration file.

## I.4 TRAINING DETAILS

**BARREL SFT and Distill SFT**  We train SFT models on the designated dataset for 2 epochs with a learning rate of $1 \times 10^{-5}$. The maximum input length is set to 1024 tokens, while the maximum output length is 2048 tokens, with a total sequence cutoff of 4096. Training is conducted with a batch size of 32, and we adopt the Adam optimizer with $\beta_1 = 0.9$ and $\beta_2 = 0.95$. All SFT training is performed using `bf16` precision and conducted using 4 NVIDIA A100 80G GPUs.

**GRPO**    We use the verl [3] framework to conduct GRPO training. For BARREL GRPO, we use our BARREL-SFT models as initialization. GRPO training is performed over 8 epochs with a learning rate of $1 \times 10^{-6}$. We set the batch size to 128 and the mini batch size to 16. The maximum prompt length is 512 tokens, and the maximum response length is 2048 tokens. We enable dynamic batch sizing and incorporate KL-divergence-based regularization with a coefficient of 0.001, using the `low_var_kl` loss type. Rollouts are conducted using the vLLM engine with a temperature of 0.6 and 8 parallel samples $n$ in GRPO. All GRPO training is conducted using 4 NVIDIA A100 80G GPUs.

## J    RANDOMNESS IN GRPO TRAINING

| Method | Accuracy (Avg.) | Truthfulness (Avg.) | Reliability (Avg.) |
|---|---|---|---|
| Distill | 38.43 | 39.33 | 39.33 |
| BARREL (reported in our paper) | 40.70 | 70.40 | 61.58 |
| BARREL (with 5 random seeds) | $41.11 \pm 1.79$ | $69.31 \pm 4.14$ | $61.44 \pm 0.97$ |

Table 12: Seed variability experiments for DeepSeek-R1-Distill-Llama-8B.

Compared to the Distill baseline, our method achieves notable gains in reliability: 1) DeepSeek-R1-Distill-Llama-8B: from 39.33% to 61.58%. 2) DeepSeek-R1-Distill-Qwen-7B: from 26.21% to 53.12%. These improvements exceed 20%, which we believe is substantial and not marginal. But it could be necessary to account for randomness:

**Fair Comparison**    We used default random seed settings and made no hard-coded decisions beyond the reward design, ensuring fair conditions across methods.

**On Random Variability**    It is widely acknowledged that randomness is inherent in LRM RL training and we could also prove our improvement is statistically significant, as shown by the seed variability experiments in Table J.

As shown, the standard deviation for reliability is 0.97, which is significantly smaller than the 20% improvement we report. Thus, the observed performance gains cannot be attributed to randomness.

We also analyzed the randomness of sampling 1k subsets from the test set and found its impact on reliability to be within 1%.

## K    ANALYSIS OF SAMPLING VARIANCE AND STABILITY

To ensure that the performance gains of BARREL are robust and not artifacts of sampling noise, we conducted a detailed analysis of output variance. A key concern in abstention-based systems is whether the decision to abstain is stable across different sampling runs. We investigated the stability of the model's decision-making process—specifically, how often the model fluctuates between answering and abstaining on the same input.

### K.1    MEASUREMENT OF INCONSISTENCY

We quantified the variance by measuring the *inconsistency rate* across $k = 4$ independent sampling runs for each question in the test set. Since the vanilla model always attempts an answer while BARREL can choose to abstain, we define inconsistency for each setting as follows:

- **Vanilla Model:** Defined as the proportion of cases where the model fluctuates between generating a *correct* and an *incorrect* response. This measures the inherent instability of the base model's knowledge retrieval.

---

[3]https://github.com/volcengine/verl

**Reasoning Examples for different models**

| ICL-IDK | Distill | Ours (SFT) | Ours (GRPO) |
|---|---|---|---|

Question: Which is the oldest of the eight Ivy League Universities, founded in 1636 as New College?

| ICL-IDK | Distill | Ours (SFT) | Ours (GRPO) |
|---|---|---|---|
| **Answer: Dartmouth** 
 **Reasoning:** Okay, so I need to figure out ... Hmm, I remember that the Ivy League is a group of top-tier universities in the US, but I'm not exactly sure about the order or the exact founding dates of each. ... Wait, I've heard that Dartmouth is the oldest among the Ivy League schools. ... Harvard is older than Dartmouth, I believe, but I'm not sure if it's part of the Ivy League. ... I think Dartmouth was founded in 1636 as New College. Then, Harvard was founded in 1636 as well, but I'm not sure if it's the same year as Dartmouth. ... So, putting it together, the oldest ... is Dartmouth. I think that's correct, but I'm a bit uncertain because I might be mixing up the exact founding dates and the order. ... I'll go with Dartmouth as the answer. | **Answer: Harvard** 
 **Reasoning:** Okay, so I need to figure out ... Let me start by recalling what I know about the Ivy League. The Ivy League ... Let me think about the founding dates of these schools. I remember that Harvard is one of the oldest. When was Harvard founded? I think it was around the 17th century. Let me check my memory. I recall that Harvard was founded in 1636. Yeah, that's right. ... So that must be the answer. But wait, are there any other schools founded around that time? ... Princeton was earlier, but wait, Princeton was actually founded in 1746, but later joined the Ivy League. ... So the answer should be Harvard. Let me just make sure there's no other school from that era. Maybe Dartmouth? No ... So the answer is Harvard. | **Answer: Harvard** 
 **Reasoning:** To determine which Ivy League university was founded in 1636 as New College, I need to consider the history and founding dates of the eight Ivy League universities. ... 1. **Harvard University**: ... , but I need to confirm the exact year. Historical records indicate that Harvard was established in 1636 as a college, which aligns with the reference answer. This makes Harvard a strong candidate for the oldest Ivy League university. 2. **Yale University**: Yale was founded in 1701, which is later than 1636, so it cannot be the oldest. This eliminates Yale as a possibility. 3. **Princeton University** ... To further verify, ... This confirms that Harvard is indeed the correct answer. | **Answer: Harvard** 
 **Reasoning:** To determine which Ivy League university was ... The first possibility that comes to mind is Harvard, as it is widely regarded as one of the oldest institutions in the United States. However, I need to verify this by considering other possible candidates and examining the evidence. 1. **Harvard University**: I know that Harvard is often associated with the founding of the Ivy League, and I recall that it was established in 1636. ... This seems like a strong candidate, but I need to confirm this further. 2. **Yale University**: ... 3. **Princeton University**: ... Now, I need to critically evaluate the evidence for Harvard. ... The other Ivy League universities were founded much later, ... After considering all possibilities ... I conclude that the answer is Harvard. |

Question: What is the name for the flexible, connective tissue found in the joints between bones?

| ICL-IDK | Distill | Ours (SFT) | Ours (GRPO) |
|---|---|---|---|
| **Answer: ligaments** 
 **Reasoning:** Okay, so I have this question about biology, specifically about the body's connective tissues. I think there's cartilage, which I know is found in places like the ends of bones, right? ... But wait, cartilage is more about cushioning and reducing friction, I believe. ... Wait, there's also something called ligaments. ... Let me make sure I'm not confusing this with tendons. Tendons connect muscles to bones, right? So they're also connective tissue, but they serve a different purpose. ... so it's more about the structure that holds the bones together, which would be ligaments. So putting it all together, the flexible, connective tissue in the joints between bones is called ligaments. I think that's the correct answer here. | **Answer: cartilage** 
 Okay, so the question is asking for the name of the flexible, connective tissue found in the joints between bones. Hmm, let's think. ... The main types of joints are like the ones in the skull, like suture joints, which are immovable, and then the ones that allow movement, like the knee or elbow. Wait, the question specifies it's connective tissue. So connective tissue is stuff like cartilage, right? Because bone is hard and dense, but the stuff between them... Oh, right! Cartilage is the flexible stuff that cushions the joints. Like in the ears or nose, those are cartilage. So in the joints, the cartilage allows movement but provides some protection against friction. So the answer should be cartilage. Let me double-check. ... So the answer is cartilage. | **Answer: cartilage** 
 **Reasoning:** To determine the name of ... I need to consider ... The first possibility that comes to mind is cartilage. ... However, I need to ensure that this is the correct term ... Another possibility is ligaments. Ligaments are also connective tissues that connect bones to other bones at joints, but they are not the tissue itself. ... Tendons are another type of connective tissue, but they connect muscles to bones and are not involved in the joint itself. ... Finally, I consider the possibility of synovial fluid, which is the fluid found in the joint cavity. ... Now, I need to verify whether cartilage is indeed the correct answer. Cartilage is the ... Ligaments and tendons do not fit the description of ... I conclude that the flexible, connective tissue found in the joints between bones is cartilage. | **Answer: cartilage** 
 **Reasoning:** To determine the name of ... I need to consider ... The first possibility that comes to mind is cartilage. ... This seems like a strong candidate, but I should consider other possibilities to ensure I am not missing anything. Another possibility could be ligament ... However, ... Another option might be synovium. ... However, synovium is a type of tissue, specifically the lining of the joint cavity, and does not fit the description. Finally, I should consider tendons. ... Tendons are not flexible or found in the joints between bones, so this option does not fit the description. Now, I need to critically evaluate the evidence for cartilage. Cartilage is ... This matches the description perfectly. After considering ... the correct answer is cartilage. |

Question: What is a network connection device that can build tables that identify addresses on each network?

| ICL-IDK | Distill | Ours (SFT) | Ours (GRPO) |
|---|---|---|---|
| **Answer: switch** 
 **Reasoning:** Okay, so I need to figure out what device is being referred to in the question. ... First, I know that network devices have various roles. Common ones include routers, switches, hubs, firewalls, modems, etc. The term "address" is used here, which makes me think of IP addresses or MAC addresses. So the device might be related to managing these addresses on a network. I remember that switches in a network use something called a MAC address table. ... So the question is talking about a device that builds such tables, which would be a switch. Switches use MAC address tables to keep track of which device is connected to which port. Therefore, the answer should be a switch. | **Answer: router** 
 **Reasoning:** Okay, so the question is asking about ... Let me think. ... But then there's a router. Routers connect different networks, like in a WAN. They use routing tables to direct packets between networks. ... Wait, the question says ön each network.So if it's on each network, maybe it's a switch because switches operate within a single network segment. Routers are more about connecting different networks. ... Alternatively, maybe the answer is a router. ... But I'm still not 100% sure. Alternatively, maybe the answer is a switch. I think the correct answer here is a router. Wait, but maybe the answer is a switch. Let me check ... I think the answer is a router. Let me go with router. | **Answer: I don't know** 
 **Reasoning:** The user is asking about a network connection device that can build tables to identify addresses on a network. ... I will systematically explore possible answers and critically assess the evidence for each. First, could it be a "router"? ... Does a router specifically "build tables" to identify addresses? ... Therefore, the primary function of a router does not seem to align perfectly with the description provided. Next, could it be a "switch"? ... They use a dynamic or static mapping of MAC addresses to IP addresses. This does not seem to match the description either. Could it be a "firewall"? ... After considering these possibilities, ... I have critically examined each candidate and found that I lack sufficient evidence to confidently identify the correct answer. | **Answer: router** 
 **Reasoning:** To answer the question, I need to ... The first possibility that comes to mind is a "router". ... They use routing tables ... which aligns perfectly with the description in the question. However, I should consider other possible devices ... Another possibility could be a "switch". ... However, switches do not maintain ... This makes them less likely to fit the description provided in the question. Another candidate might be a "firewall". ... This makes them a less likely match. ... To further evaluate the evidence for each: - Routers are widely used in ... - Switches are ..., making them less relevant. - Firewalls are ..., making them a weaker fit. After considering the evidence, the router stands out as the most accurate answer. |

Table 10: Additional Reasoning path examples of different methods on DeepSeek-R1-Distill-Llama-8B. We colored with wrong factual thinking leading to **wrong answers**, factual thinking alerting **not sure** and the good factual thinking leading to **correct answers**.

**Reasoning Examples for different models**

| ICL-IDK | Distill | Ours (SFT) | Ours (GRPO) |
|---|---|---|---|

Question: What do methanogens act as in sewage treatment plants?

| ICL-IDK | Distill | Ours (SFT) | Ours (GRPO) |
|---|---|---|---|
| **Answer: Biogas producers** **Reasoning:** Okay, so I need to figure out ... I remember that methanogens are types of bacteria, but I'm not exactly sure about their role in sewage ... I recall something about biogas being produced ... So perhaps methanogens convert the organic waste into biogas ... Alternatively, maybe they're acting as decomposers ... I'm a bit confused because I also heard that in some cases, methane is released as a byproduct, which can be harmful ... I'm pretty sure that's the case, but I'm not 100% certain. I think I'll go with that. Methanogens in sewage treatment plants act as biogas producers ... | **Answer: methane** **Reasoning:** Okay, so the question is asking ... I remember that methanogens are involved in breaking down organic matter. ... Maybe methanogens are responsible for converting certain compounds. ... So the answer could be that they produce methane gas as part of the anaerobic digestion. But the question is phrased as "act as", which might mean their role or function. ... So maybe the answer is that they produce methane gas ... Let me check my notes. ... Let me confirm. ... So the answer is methane. ... But the question says "act as", so maybe the answer is methane. ... Yeah, I think that's right. | **Answer: I don't know** **Reasoning:** The user is asking ... First, I consider the possibility ... act as decomposers ... However, what specific evidence supports this? ... But are they exclusively decomposers? ... I need to investigate further. Next, I consider ... act as aerobes. ... Do I have any evidence that ... However, I cannot entirely rule out the possibility that I need to challenge this assumption. Another possibility is ... act as catalysts ... Do I have any evidence to definitively say they act as catalysts ... Finally, I consider ... act as nitrogen-fixers ... Is there evidence that ... so this seems less likely. After evaluating these possibilities, I find that ... However, I lack definitive evidence to ... I cannot confidently ... | **Answer: decomposers** **Reasoning:** To answer the question ... The first possibility that comes to mind is decomposers. ... I need to verify this ... Another possibility is reducers. However, ... This suggests that "reducers" is not the correct answer. A third possibility is filters. However, ... This rules out "filters" as a plausible answer. Next, I need to evaluate the evidence for "decomposers." ... This strongly supports the idea ... To challenge this assumption, I must ask: ... While ... This suggests that "decomposers" is the most accurate answer. Finally, I verify this conclusion by considering ... This confirms that their role as decomposers. |

Question: What team does controversial quarterback Michael Vick play for in the National Football League in the USA?

| ICL-IDK | Distill | Ours (SFT) | Ours (GRPO) |
|---|---|---|---|
| **Answer: Cleveland Browns** **Reasoning:** Okay, so I need to figure out ... But I'm not entirely sure about ... I think he was with the Cleveland Browns ... But I'm not 100% sure. I should double-check that. ... I think he might have played for the Cleveland Browns during the early 2000s. Maybe around 2003 or so. ... Wait, was it the Cleveland Browns or the Philadelphia Eagles? I think he was with the Browns when he got suspended by the Eagles. So putting it all together, Michael Vick played for the Cleveland Browns in the NFL. I think that's the correct answer. | **Answer: Miami Dolphins** **Reasoning:** Okay, so I need to figure out ... I think he started with the Philadelphia Eagles. Wait, but then he was traded to the Cleveland Browns, right? ... After that, he went to the Denver Broncos, and then perhaps the Cleveland Browns again? Or maybe ... ? ... I think the last team he played for was the Miami Dolphins. But I should confirm. ... Wait, another angle: his draft year. ... So the answer should be Miami Dolphins. ... But wait, maybe he was traded again after Miami? I don't recall him ... So the answer is Miami Dolphins. | **Answer: Philadelphia Eagles** **Reasoning:** To determine ... The most ... that comes to mind is the Philadelphia Eagles. ... However, I need to verify this and consider other possibilities. Other teams that might be relevant include: ... Evaluate evidence for each possibility 1. Miami Dolphins: ... but there is no evidence to suggest ... 2. Denver Broncos: ... but the Broncos are not the final team he ended up with. 3. Philadelphia Eagles: ... This is the team he ended up playing for after the trade. ... Challenge assumptions. Could I be misinterpreting the evidence? ... No, the evidence overwhelmingly points to the Eagles ... Verify the evidence ... This is consistent with the timeline of his NFL career. | **Answer: Eagles** **Reasoning:** To determine ... The first possibility that comes to mind is the Philadelphia Eagles. ... but I need to verify this. ... Another possibility is the Miami Dolphins ... However, I don't recall any specific evidence ... but I need to keep it in mind for further verification. A third possibility is the Dallas Cowboys ... However, I don't have any specific evidence ... To resolve this, I need to focus on concrete evidence. ... This aligns with my initial thought. Additionally, the Eagles ... Next, I should challenge this assumption. ... To verify, I check reliable sources like ... Finally, I apply adversarial self-critique. ... The evidence is sufficient ... |

Question: Who created Rumpole of the Bailey?

| ICL-IDK | Distill | Ours (SFT) | Ours (GRPO) |
|---|---|---|---|
| **Answer: Andrew Marquardt** **Reasoning:** Okay, so I need to figure out ... Wait, isn't that by Andrew Marquardt? Or was it someone else? I'm a bit confused ... Maybe it's by David Suchet? Wait, no, that's the actor who played Rumpole. ... Let me think about the name again. The play is by Andrew Marquardt, right? But wait, I'm not entirely sure. I think I should check ... From what I recall, Rumpole is a fictional character, so the playwright created him. Therefore, the creator would be Andrew Marquardt. I'm a bit unsure ... but I think in this case, it's the playwright who created the character. So I'll go with Andrew Marquardt as the creator. | **Answer: Charles Dickens** **Reasoning:** Okay, so I need to figure out ... I know that Charles Dickens wrote a lot of classic stories. Rumpole of the Bailey... I think that's from "Oliver Twist." Wait, no, Rumpole is the character in "Oliver Twist," right? ... But was Rumpole a fictional character created by Dickens? ... Wait, maybe the answer is Charles Dickens. ... But I should make sure. Let me recall. ... So Dickens created Rumpole of the Bailey. ... Dickens often created his own characters, so yes, he created Rumpole. But maybe there's a historical figure with that name? Let me check my memory. Charles Dickens wrote "Oliver Twist," and Rumpole is the friend of Oliver. So the answer should be Charles Dickens. | **Answer: I don't know** **Reasoning:** The user is asking about ... First, I consider whether the creator could be a famous British comedian ... However, I must ask myself: Do I have any specific recollection or evidence ...? No, I don't have any concrete evidence or memory of ... Next, I consider whether the creator might be a lesser-known figure ... Again, I must challenge this assumption: ... No, I don't have any such evidence. ... Finally, I consider whether the creator might be ... However, I must ask myself: Do I have any evidence ... No, I don't have any such evidence. I have systematically examined multiple possibilities. ... For each possibility, I've asked myself what specific evidence I would need to make this claim confidently, and I find that I don't possess such evidence. | **Answer: I don't know** **Reasoning:** The user is asking about ... First, I need to consider whether the show was created by ... What evidence supports this? I don't have any direct evidence ... Next, could it have been created by ... However, I don't have any specific information ... This possibility also seems unsupported. Could it have been created by a more recent ... ? However, I still lack specific evidence ... Without concrete evidence, I cannot confidently assert this as the answer. ... I have systematically examined multiple possibilities. ... For each possibility, I have asked what specific evidence I would need to make this claim confidently, and I find that I don't possess such evidence. |

Table 11: Reasoning path examples of different methods on DeepSeek-R1-Distill-Qwen-7B. We colored with wrong factual thinking leading to **wrong answers**, factual thinking alerting **not sure** and the good factual thinking leading to **correct answers**.

- **BARREL:** Defined as the proportion of cases where the model fluctuates between *answering* and *abstaining*. This measures the stability of our proposed tuning method's decision boundary.

## K.2 QUANTITATIVE RESULTS

Table 13 presents the comparison of inconsistency rates across different backbone models. Contrary to the concern that an additional decision module might introduce instability, BARREL consistently exhibits lower variance compared to the vanilla baselines.

| Backbone Model | Vanilla Inconsistency | BARREL Inconsistency |
|---|---|---|
| DeepSeek-Distill-Llama-8B | 31.93 | **19.43** |
| DeepSeek-Distill-Qwen-7B | 23.13 | **17.13** |
| Qwen3-8B | 18.97 | **14.87** |

Table 13: Comparison of output inconsistency rates (%) across 4 independent samples. Lower values indicate higher stability. BARREL demonstrates greater consistency in its decision to abstain than the vanilla model shows in its correctness.

For example, on the DeepSeek-Distill-Llama backbone, the inconsistency rate drops significantly from 31.93% (Vanilla) to 19.43% (BARREL). This indicates that BARREL does not amplify instability; rather, it stabilizes the output by effectively masking uncertain predictions that are prone to fluctuation in the vanilla model.

## L    DETAILED EXAMPLES

We provide more detailed examples on DeepSeek-R1-Distill-Llama-8B and DeepSeek-R1-Distill-Qwen-7B in Table 10 and Table 11. In addition to the two main thinking patterns in current LRMs—second-thought spiraling and last-minute guessing—introduced earlier in this paper, we also observed an additional pattern on DeepSeek-R1-Distill-Qwen-7B that sometimes results in incorrect answers where the model initially proposes an incorrect answer and subsequently engages in multiple rounds of self-checking. However, these self-checks are characterized by expressions of uncertainty and lack of firm commitment. Ultimately, the model still adheres to the original incorrect answer. However, this flawed reasoning pattern can also be effectively transformed into a more reliable and coherent one through our BARREL training framework.

## M    ADDITIONAL DETAILS

We provide the links and licenses of the datasets and code used in our paper as follows:

**Code**    We conduct SFT using our own codebase, which is built on top of the Transformers library[4] and DeepSpeed[5]. And we conduct GRPO training using the verl framework [6].

**Data**    We make use of the following publicly available datasets. (1) *TriviaQA:* Open-domain question–answering corpus drawn from Wikipedia and the web (Apache 2.0 License)[7];

(2) *SciQ:* 13 679 multiple-choice science questions spanning physics, chemistry, biology, and more (CC BY-NC 3.0 License)[8];

(3) *NQ-Open:* Open-domain variant of Natural Questions covering real Google queries (CC BY-SA 3.0 License)[9];

---

[4]https://github.com/huggingface/transformers
[5]https://github.com/deepspeedai/DeepSpeed
[6]https://github.com/volcengine/verl
[7]https://github.com/mandarjoshi90/triviaqa
[8]https://huggingface.co/datasets/allenai/sciq
[9]https://github.com/efficientqa/nq-open

(4) *SimpleQA:* Complex factuality benchmark (MIT License)[10];

(5) *MATH-500:* 500-problem subset of the MATH benchmark for compact maths evaluation (MIT License)[11];

(6) *MATH:* full-scale mathematics problem benchmark (MIT License)[12].

(7) *SelfAware:* unanswerable questions (Apache 2.0 License) [13].

## N   MODELS USED IN OUR EXPERIMENTS

We provide the download links to the models used in our experiments as follows:

- DeepSeek-R1-Distill-Llama-8B      (`https://huggingface.co/deepseek-ai/DeepSeek-R1-Distill-Llama-8B`)
- DeepSeek-R1-Distill-Qwen-7B      (`https://huggingface.co/deepseek-ai/DeepSeek-R1-Distill-Qwen-7B`)
- DeepSeek-R1-Distill-Qwen-32B      (`https://huggingface.co/deepseek-ai/DeepSeek-R1-Distill-Qwen-32B`)
- DeepSeek-R1-Distill-Llama-70B      (`https://huggingface.co/deepseek-ai/DeepSeek-R1-Distill-Llama-70B`)
- DeepSeek-R1 (`https://huggingface.co/deepseek-ai/DeepSeek-R1`)
- QwQ-32B (`https://huggingface.co/Qwen/QwQ-32B`)
- Qwen3-8B (`https://huggingface.co/Qwen/Qwen3-8B`)

## O   QUICK ANALYSIS OF THE UNDERLYING MECHANISM

Previous work (Liang et al., 2024) has shown that using the hidden states of LLMs to build a classifier can achieve high consistency with the sampling strategy mentioned in Section 3.1. This suggests that the model has the potential to recognize knowledge boundaries and to say "sorry" when faced with unknown questions. In our approach, however, we boost the ability to identify these boundaries through an explicit reasoning process.

## P   DISCUSSION ON DOMAIN GENERALIZATION

While our primary experiments focus on Factual QA, the proposed BARREL framework establishes a principled training paradigm designed for broader applicability. The core methodology—utilizing SFT to seed refusal behaviors and GRPO to instill calibrated abstention incentives—is agnostic to the specific domain. However, adapting this framework requires different patterns of "reliable reasoning" tailored to the specific failure modes of different tasks.

**Reliability and the Nature of "Overthinking".**    In the factual domain, we identify "overthinking" (generating extensive rationales for unknown facts) as a primary symptom of unreliability. Consequently, our current implementation specifically targets this behavior. However, this symptom does not apply for all domains. For instance, our preliminary analysis on instruction-following tasks (e.g., IFEval) reveals minimal difference in thought chain length between correct and incorrect responses (e.g., 496 vs. 501 tokens for DeepSeek-R1-Distill-Llama-8B), suggesting that lengthy reasoning is not pathological for reliability issue.

---

[10]`https://github.com/openai/simple-evals`
[11]`https://huggingface.co/datasets/HuggingFaceH4/MATH-500`
[12]`https://github.com/hendrycks/math`
[13]https://github.com/yinzhangyue/SelfAware

**Adapting BARREL to Reasoning Domains.** Adapting BARREL framework to other domains is still an exciting open question. To generalize BARREL to mathematics or complex instruction following, the construction of the SFT refusal data and the calibration of GRPO rewards should shift to modeling domain-specific boundary patterns:

- **Instruction Following:** Reliability in this domain is tied to constraint satisfaction and execution planning rather than fact retrieval. A reliable reasoning pattern should involve simulating execution steps and proactively detecting potential constraint violations. The SFT phase would thus focus on demonstrating "simulation-and-check" behaviors before committing to an answer.

- **Mathematics:** Mathematical tasks involve complex logical chains where errors propagate. Here, mitigating unreliability requires sensitivity to intermediate reasoning flaws. A reliable process entails detecting when a logical step becomes uncertain, interrupting the flawed chain, and transitioning to an expression of uncertainty. The GRPO reward modeling would explicitly penalize the completion of hallucinated derivations while rewarding early error detection.

In summary, while the specific manifestations of reliable reasoning differ across domains, BARREL provides the foundational infrastructure to verify LRMs' ability to reason reliably. Future work can leverage this framework by formalizing domain-specific reliable reasoning patterns to construct targeted SFT data and calibrated reward signals.

## Q LIMITATIONS

Although we have evaluated and validated the performance of our method, restricted by our limited computing resource, we mainly utilize DeepSeek-R1-Distill-Llama-8B, DeepSeek-R1-Distill-Qwen-7B and Qwen3-8B to perform our study, which are relatively small LRMs.

In our experiments, though we have covered different datasets, we mainly utilizing verifiable questions, which could be evaluated at test time and reward at training time using string matching method. How to teach LRMs to learn knowledge boundary and behave more deliberatively on open-end questions, like writing articles or providing opinions, remains a valuable topic for future work.

We adopted the general acknowledged sampling strategy to annotate question types, but we acknowledge that this proxy has inherent limitations. For instance, when the model is genuinely confused between two specific options (e.g., oscillating between distinct answers due to uncertainty), the current consistency-based metric might fail to capture this state accurately, potentially introducing errors. As a result, there is a certain proportion of mislabeled data in the SFT stage. Our subsequent adaptive GRPO training will help alleviate this problem, but we need to emphasize that constructing better proxies that can distinguish between ignorance and genuine confusion remains an open challenge, and it is difficult to obtain a robust model directly through SFT. Detecting issues in the reasoning process remains an open question. If an accurate method for monitoring reasoning could be developed, both false positives and true negatives could be reduced to some extent.

As shown in Appendix K, although BARREL improves stability over vanilla baselines, it still exhibits an inconsistency rate of approximately 20% across sampling runs. This indicates that for a subset of borderline queries, the decision to abstain remains sensitive to sampling noise, suggesting room for further improvement in calibration stability.

In the paper, we mainly focus on whether the model can reason to claim uncertainty through Reinforcement Learning with Verifiable Rewards (RLVR), without introducing methods such as CoT monitoring and detection. Building detectors for CoT hallucinations and uncertainty is an important future direction, which goes beyond the scope of our current work.

Another limitation lies in the reward design. Currently, we employ a fixed rejection reward, which implicitly ties the optimal policy to a fixed internal confidence threshold in limited training steps. This could limit the model's flexibility for scenarios requiring different risk tolerances (e.g., higher or lower refusal rates). Future work could explore conditioning the model on a dynamic rejection reward specified in the prompt, enabling the model to generalize to multiple thresholds without retraining.

## R    BOARDER IMPACT

Although existing LRMs have demonstrated strong capabilities, hallucinations frequently occur in real-world applications, raising doubts about their reliability. Our work enhances the reliability of LLM responses by teaching the models to express uncertainty and refuse to answer when appropriate. As mentioned in Appendix P, future work can leverage BARREL framework by formalizing domain-specific reliable reasoning patterns to construct reliable LRMs for different domains. We hope our work will inspire more research in this area to further improve model reliability, enabling humans to place greater trust in the positive responses provided by LRMs.

## S    THE USE OF LARGE LANGUAGE MODELS

In this paper, we utilized LLMs to perform grammatical error checking, remove redundancies, and refine the vocabulary in various sections of the paper. It is important to note that the LLM's role was limited to polishing and checking the text, not to writing any paragraphs from scratch.

