# OpenReview forum: "BARREL: Boundary-Aware Reasoning for Factual and Reliable LRMs"
_ICLR.cc/2026/Conference — ICLR 2026 Poster_

### Official Review · Reviewer_2MeN · 2025-10-27

**Soundness:** 2
**Presentation:** 2
**Contribution:** 3
**Rating:** 4
**Confidence:** 4

**Summary:**

This paper presents BARREL, a training paradigm consisting of a SFT and RL stage with the goal of making models aware of their factual knowledge boundaries by learning to say “I don’t know” when they are unsure. In contrast to the standard RL paradigm, BARREL assigns an abstention reward if the model outputs “I don’t know”, with the magnitude of this reward being between the rewards for a correct and incorrect answer. Results show that BARREL maintains competitive overall accuracy in factual domains while significantly increasing precision (when models choose to output an answer, their accuracy is much higher). Ablations highlight the importance of both SFT and RL stages, as well as the significance of the proposed reward.

**Strengths:**

- The paper is well-motivated. Hallucinations are a major problem with LLMs and making models aware of their factual boundaries is an an important problem. The paper also correctly identifies that the standard paradigm does not reward models to express their uncertainty.

- The paper is generally well-written and has a decent flow to it.

- The results are impressive. Models maintain accuracy while also learning to abstain.

**Weaknesses:**

- **Proxy Definition**: The "*knowledge labeling proxy*" (classifying a question as "known to the model") is a weak indicator of model knowledge. It fails to account for uncertainty when multiple plausible answers exist. Say a question has 10 possible answers. The model might not know the correct answer, but has some sampling probability for each of the answers. The proposed framework would classify this question as something known to the model, which seems strange. A majority-vote–based proxy might align more closely with actual model uncertainty.
- **Threshold-based Reward**: The rule-based reward design implicitly fixes a single confidence threshold - that is, the model should only answer the question if its confidence is greater than $\alpha$:

    $\alpha \ge \frac{r_s - r_w}{r_c - r_w}.$

    This can be computed by writing the expected reward for the model for different correctness probabilities and checking if it is greater than the abstention reward. A limitation of this framework is that it removes flexibility across use cases — for instance, scenarios requiring high confidence (e.g., 99\%) versus those tolerant of lower confidence (e.g., 70\%) cannot be easily tuned.

- **Limited Related Work**: The related work section omits several recent RL-based approaches that explicitly train models to reason about their uncertainty [1,2,3].

- **Experimental Design**: Some important baselines such as classifier and vanilla RLVR are missing from the main paper (see questions below).

[1]: Stangel, P., Bani-Harouni, D., Pellegrini, C., Özsoy, E., Zaripova, K., Keicher, M., & Navab, N. (2025). Rewarding Doubt: A Reinforcement Learning Approach to Calibrated Confidence Expression of Large Language Models. arXiv preprint arXiv:2503.02623.

[2]: Xu, T., Wu, S., Diao, S., Liu, X., Wang, X., Chen, Y., & Gao, J. (2024). Sayself: Teaching llms to express confidence with self-reflective rationales. arXiv preprint arXiv:2405.20974.

[3]: Damani, M., Puri, I., Slocum, S., Shenfeld, I., Choshen, L., Kim, Y., & Andreas, J. (2025). Beyond binary rewards: Training lms to reason about their uncertainty. arXiv preprint arXiv:2507.16806.

**Questions:**

- What value of $\alpha$ do the authors’ chosen rewards correspond to, and how was this default confidence threshold determined? (I think it corresponds to 25%)
- The authors mention (lines 356-360) “*it is worth noting the accuracy improvement caused by increasing the ratio does not reflect an actual improvement in model capability—it merely reduces the number of incorrect refusals on known questions.*” If this is the case, then why does the ICL baseline not match the accuracy of BARREL? This hypothesis does not seem to be correct.
 - What is the motivation behind the reliability metric — what does it intuitively measure or indicate?
 - How much variance is observed in BARREL’s outputs due to sampling? Specifically, how often does the model sometimes answer and sometimes abstain on the same question?
- Could the authors include the abstain rate $\left( \frac{N_w}{N} \right)$ in the main results table for better interpretability?
- **Probing/Classifier baseline** should also be added to the paper. Train a classifier/probe on the vanilla GRPO (no truthful rewards) outputs to predict the probability that the answer is correct. This probe can then be used to manually replace answers with “idk”.
- "*Quick Analysis of the Underlying Mechanism*": This is not an experiment and should be presented in a different way.
- "*Did GRPO sacrifice pass@k for better pass@1*": I do not think enough evidence has been provided to support this claim. The gap between SFT and GRPO seems to reduce significantly, suggesting that SFT has better scaling performance.
- It will be useful to ablate the importance of SFT and the impact of RL with/without idk rewards. I think most of these results are already in the appendix, but some of the numbers should be added to the main results table. In particular, RL (without rejection rewards) should be added to the table as that is a very standard baseline.

---

> ### Author Response · Authors · 2025-11-20
> **Response by Authors (1/5)**
>
> We sincerely thank the reviewer for the detailed and thoughtful feedback. We appreciate the recognition of the paper’s strong motivation, clear writing, and comprehensive experimental results. For clarity and ease of understanding, we grouped the reviewer's weaknesses and questions into four major categories and addressed them accordingly.
>
> **Part 1: Metrics, Settings, and Assumptions**
>
> We appreciate the reviewers' detailed question of our metric definitions, proxy assumptions, and reward design choices. We provide our feedbacks as follows.
>
> > **[Q1]**:  Proxy Definition: The "knowledge labeling proxy" (classifying a question as "known to the model") is a weak indicator of model knowledge. It fails to account for uncertainty when multiple plausible answers exist. Say a question has 10 possible answers. The model might not know the correct answer, but has some sampling probability for each of the answers. The proposed framework would classify this question as something known to the model, which seems strange. A majority-vote–based proxy might align more closely with actual model uncertainty. (Weakness 1)
>
> **[R1]**: Our approach is grounded in well-established practices in factual QA evaluation, as supported by prior work [1][2][3]. In factual QA datasets, the task is not open-ended—each question corresponds to a single correct answer. The only complication is that a question may have multiple valid surface forms of that same answer, and the datasets themselves already account for this.
>
> For example, in TriviaQA, consider the question: "A colony of Britain until 1956, with the capital city of Khartoum, the south of what African country became an independent state in July 2011?"
>
> The gold answer list includes forms such as "government of sudan,” "republic of sudan,” "northern sudan,” "السودان,” "sudan,” and others. These entries all refer to the same underlying answer: Sudan.
>
> [1] Gekhman, Z., Yona, G., Aharoni, R., Eyal, M., Feder, A., Reichart, R., & Herzig, J. (2024). Does fine-tuning LLMs on new knowledge encourage hallucinations? EMNLP 2024.
>
> [2] Xue, B., Mi, F., Zhu, Q., Wang, H., Wang, R., Wang, S., Yu, E., Hu, X., & Wong, K.-F. (2025). UAlign: Leveraging uncertainty estimations for factuality alignment on large language models. ACL 2025.
>
> [3] Li, M., Zhao, Y., Zhang, W., Li, S., Xie, W., Ng, S.-K., Chua, T.-S., & Deng, Y. (2025). Knowledge boundary of large language models: A survey. ACL 2025.
>
> > **[Q2]**: What is the motivation behind the reliability metric — what does it intuitively measure or indicate? (Question 3)
>
> **[R2]**: The motivation for introducing the reliability metric is to consider both helpfulness and truthfulness. As mentioned in Line 288, truthfulness metric can be trivially optimized by refusing to answer, so it fails to distinguish responsible abstention from indiscriminate avoidance. Reliability instead measures the balance between (1) giving correct answers when the model chooses to respond and (2) abstaining appropriately when uncertain.
>
> When the answer rate is low, we encourage the model not to reflexively refuse but rather to attempt to provide assistance. Conversely, when the answer rate is high, we believe the model should become more cautious to avoid errors. This metric balances the model's helpfulness and truthfulness while mitigating the risks of it becoming overly conservative or excessively aggressive. Thus it makes reliability a more faithful indicator of real-world performance than truthfulness alone, consistent with prior work [4] [5].
>
> [4] Xu, H., Zhu, Z., Zhang, S., Ma, D., Fan, S., Chen, L., & Yu, K. (2024). Rejection improves reliability: Training LLMs to refuse unknown questions using RL from knowledge feedback. COLM 2024
>
> [5] Kim, H. J., Kim, Y., Lee, S., & Kim, T. (2025). When to speak, when to abstain: Contrastive decoding with abstention. ACL 2025.

---

> ### Author Response · Authors · 2025-11-20
> **Response by Authors (2/5)**
>
> > **[Q3]**: Threshold-based Reward. The rule-based reward design implicitly fixes a single confidence threshold — that is, the model should only answer the question if its confidence is greater than α: \alpha \ge \frac{r_s - r_w}{r_c - r_w}.This can be computed by writing the expected reward for the model for different correctness probabilities and checking if it is greater than the abstention reward. A limitation of this framework is that it removes flexibility across use cases — for instance, scenarios requiring high confidence (e.g., 99%) versus those tolerant of lower confidence (e.g., 70%) cannot be easily tuned. What value of  do the authors' chosen rewards correspond to, and how was this default confidence threshold determined? (I think it corresponds to 25%) (Weakness 2 and Question 1)
>
> **[R3]**: The static threshold formula does not apply because GRPO optimizes rewards relative to a dynamic baseline, not absolute values. While the reviewer’s threshold calculation, $\alpha = (r_s - r_w) / (r_c - r_w)$, is correct for a classical decision framework, GRPO does not optimize against absolute reward values but instead uses normalized rewards relative to a dynamic group-averaged baseline (Iine 247-249). So theoraticly there is no fixed convergence point or static 25% confidence threshold.
>
> When the model exhibits both refusal and correct-answer behaviors simultaneously, our framework incentivizes the model to reason its way to the correct answer. For example, in a group of 8 rollouts with 1 Correct ($+1.0$), 6 Incorrect ($-1.0$), and 1 Refusal ($-0.5$), the group average reward $\overline{R}$ becomes $\overline{R} = \frac{(1 \cdot 1) + (6 \cdot -1) + (1 \cdot -0.5)}{8} = -0.6875$. So here the final reward for correct answer is $+1.6875$ and the reward for refusal is $+0.1875$. It's clear that the model will be optimized to give the correct answer.
>
> It is also worth noting that the framework allows different values of $r_s$ to be assigned to different subsets of data. While the choice of $r_s$ does affect the accuracy–truthfulness trade-off within a fixed training budget as also illustrated in Figure 5, we do not consider this a limitation of the framework; rather, it reflects its flexibility in allowing different $r_s$ values for different tasks. Our framework provides a useful degree of freedom. Practitioners can tune $r_s$ on their own multiple downstream datasets to match the reliability profile required by their application. Different settings naturally call for different refusal rewards: for example, factual QA may adopt values similar to those used in our experiments, whereas high-stakes domains such as medical QA may select a larger $r_s$ to encourage the model to answer only when it holds very high confidence. This flexibility is by design, enabling developers to easily adapt the framework to domain-specific risk tolerances.
>
> This discussion provides meaningful and constructive perspectives, and we will include analysis above in the final version.

---

> ### Author Response · Authors · 2025-11-20
> **Response by Authors (3/5)**
>
> **Part 2: Interpretation and Reliability of Analysis**
>
> In this section, we will address potential misunderstandings, and provide additional analysis to improve the reliability of our claims.
>
> > **[Q4]**: The authors mention (lines 356-360) "it is worth noting the accuracy improvement caused by increasing the ratio does not reflect an actual improvement in model capability—it merely reduces the number of incorrect refusals on known questions.” If this is the case, then why does the ICL baseline not match the accuracy of BARREL? This hypothesis does not seem to be correct. (Question 4)
>
> **[R4]**: We appreciate the opportunity to clarify this distinction. The confusion likely stems from comparing two different mechanisms of improvement.
>
> - SFT (BARREL) vs. ICL: The superiority of BARREL over the ICL baseline arises because SFT fundamentally corrects "pathological overthinking" patterns that ICL cannot resolve. This provides the initial jump in accuracy and capability.
>
> - Data Ratio Impact (The Quote): The specific claim on lines 356-360 refers only to the marginal gain achieved by tuning the data ratio (e.g., from $1:1$ to $3:1$) within the SFT stage.
>
> Our analysis shows that of the $9.0\%$ accuracy gain observed when increasing this ratio, $8.9\%$ resulted solely from converting incorrect refusals into correct answers (overly conservative), while only $0.1\%$ came from correcting previously wrong reasoning (capability). Thus, while SFT improves capability over ICL, adjusting the data ratio primarily optimizes the trade-off between refusal and answering, rather than teaching new reasoning skills.
>
> > **[Q5]**: "Did GRPO sacrifice pass@k for better pass@1": I do not think enough evidence has been provided to support this claim. The gap between SFT and GRPO seems to reduce significantly, suggesting that SFT has better scaling performance. (Question 8)
>
> **[R5]**: We appreciate the reviewer's careful inspection of the Pass@k results. We would like to clarify that the term "sacrifice" in this context refers to whether our GRPO stage causes absolute performance (Pass@k) to drop below that of the SFT model. (In such comparisons, we generally look at absolute values rather than relative improvements.) As cited in line 447, prior work shows that RL training on math tasks may significantly reduce pass@k, indicating a substantial loss of diversity in exchange for higher pass@1 performance.
>
> GRPO clearly outperforms SFT in accuracy, truthfulness and reliability. So our concern is that whether on pass@k factual test scenario (like pass@4), SFT is better than GRPO. We actually find that for factual QA tasks, the GRPO model does not underperform the SFT model on pass@k, demonstrating that it does not severely sacrifice sampling diversity just to improve pass@1. We will add a clarification on the definition of "sacrifice" in the revised manuscript to prevent this misunderstanding.

---

> ### Author Response · Authors · 2025-11-20
> **Response by Authors (4/5)**
>
> **Part 3: Organization and Presentation of the Manuscript**
>
> We thank the reviewers for pointing out areas where the manuscript's organization and presentation could be improved. We agree that clearer structuring and framing of certain analyses would enhance readability, and we respond to each suggestion in detail below.
>
> > **[Q6]**:Limited Related Work: The related work section omits several recent RL-based approaches that explicitly train models to reason about their uncertainty (Weakness 3)
>
> **[R6]**: We thank the reviewer for suggesting these relevant references. We will expand our related work section to include them in the final revision.
>
> 1. RL-based uncertainty approaches (First two papers): Our current draft discusses R-tuning and RLKF as representative methods for training models with refusal signals. We agree the suggested papers are relevant; however, like R-tuning/RLKF, they primarily focus on standard LLMs. Our work distinguishes itself by targeting Large Reasoning Models (LRMs) under the Long-CoT paradigm, dealing with specific refusal and safety dynamics that standard LLM methods do not address.
>
> 2. Concurrent work (Third paper): The third paper is a concurrent study which is released on ArXiv within the last two months [1]. While valuable, its methodology differs significantly from ours: it explores training a reliable reasoning model from scratch, whereas our work focuses on diagnosing failure modes in existing LRMs and correcting them through targeted training adjustments. We will discuss this distinct but complementary line of work in our revision.
>
> [1] https://iclr.cc/Conferences/2026/ReviewerGuide
>
>
>
> > **[Q7]**:Could the authors include the abstain rate  in the main results table for better interpretability? (Question 5)
>
> **[R7]**:Thank you for the suggestion. Since the abstain rate is actually truthfulness − accuracy, we previously omitted it from the main figure due to redundancy and space constraints. We will add the average abstain rate for each method in Table 1 in the final version.
>
>
>
> > **[Q8]**:"Quick Analysis of the Underlying Mechanism": This is not an experiment and should be presented in a different way. (Question 7)
>
> **[R8]**: We appreciate the suggestion. We can move this section to the appendix and add a brief pointer at line 342 directing readers to the appendix for details.
>
>
>
> > **[Q9]**:It will be useful to ablate the importance of SFT and the impact of RL with/without idk rewards. I think most of these results are already in the appendix, but some of the numbers should be added to the main results table. In particular, RL (without rejection rewards) should be added to the table as that is a very standard baseline. (Question 9)
>
> **[R9]**: Thank you for the helpful recommendation. We originally moved these results to the appendix due to page limitations. For the final version, where we have an additional page, we will move Table 5, Figure 8, and the corresponding discussion from Appendix C into the main paper.

---

> ### Author Response · Authors · 2025-11-20
> **Response by Authors (5/5)**
>
> **Part 4: Additional Experimental Analyses**
>
> We appreciate the requests for additional ablations and expanded reporting to improve the empirical completeness of our work. In this section, we respond to the question about sampling and probing baseline comparisons.
>
> > **[Q10]**: How much variance is observed in BARREL's outputs due to sampling? Specifically, how often does the model sometimes answer and sometimes abstain on the same question? (Question 4)
>
> **[R10]**: While sampling variance exists, BARREL actually exhibits lower variance than the vanilla model, indicating that our method does not amplify instability. To quantify this, we measured the inconsistency rate across 4 independent samples for each question. BARREL: The proportion of cases where the model fluctuates between answering and abstaining; Vanilla Model: The proportion of cases where the model fluctuates between correct and incorrect responses (as it does not abstain).
>
> As shown below, BARREL is consistently more stable than the vanilla baseline:
>
> | samples=4 | Vanilla Model | BARREL |
> | :--- | :--- | :--- |
> | DeepSeek-Distill-Llama | 31.93 | 19.43 |
> | DeepSeek-Distill-Qwen | 23.13 | 17.13 |
> | Qwen3 | 18.97 | 14.87 |
>
> Furthermore, our evaluation is statistically robust. On the 3,000-sample test set, the standard deviation across independent sampling runs remains below 1%. This stability holds across different data splits, confirming that our reported gains are not artifacts of sampling noise. The above analysis and discussions will be included in the final version of the paper.
>
> > **[Q11]**: Probing/Classifier baseline should also be added to the paper. Train a classifier/probe on the vanilla GRPO (no truthful rewards) outputs to predict the probability that the answer is correct. This probe can then be used to manually replace answers with "idk”(Question 6)
>
> **[R11]**: Thanks for the helpful suggestion. Following the reviewer's recommendation, we incorporated probing-style baselines that estimate the correctness of model outputs and use these estimates to decide when to abstain by answering "I don't know.” Specifically, we implemented two approaches—Verbal Confidence and Probing—and applied them to vanilla GRPO (without our abstention rewards).
>
> 1. **Verbal Confidence**: We prompt the model to explicitly outputs its confidence, which prior work has shown to be an effective and well-caliberated way [7].
>
> 2. **Probing**: Using intermediate model activations, we train a lightweight classifier to predict the probability that the model's answer is correct, inspired by recent work demonstrating that hidden representations encode rich signals correlated with factuality and error detection [8].
>
> For each method, we replace an answer with "I don't know” whenever the predicted probability is below a tuned threshold. Higher thresholds improve truthfulness but generally reduce accuracy, while lower thresholds tend to have limited effect. We selected thresholds that yielded the best trade-off in validation: 0.8 for Verbal Confidence and 0.4 for Probing. The results are shown below:
>
> |   Methods / Models | Acc (Avg.) | Truth (Avg.) | Rel (Avg.) |
> | :----: | :----: | :----: | :---: |
> | **DeepSeek-R1-Distill-Llama-8B** |  /  | / | / |
> |  ICL   |   28.13    |    28.33     |   28.33    |
> |    Vanilla GRPO + Verbal Conf    |   37.50    |    54.80     |   51.81    |
> |      Vanilla GRPO + Probing      |   40.20    |    58.77     |   55.32    |
> |              BARREL              | **40.70**  |  **70.40**   | **61.58**  |
> | **DeepSeek-R1-Distill-Qwen-7B**  | /  |  /   |  /  |
> |               ICL       |   18.07    |    18.80     |   18.79    |
> |    Vanilla GRPO + Verbal Conf    |   24.43    |    31.03     |   30.06    |
> |      Vanilla GRPO + Probing      |   18.27    |    60.30     |   42.63    |
> |              BARREL              | **28.27**  |  **74.50**   | **53.12**  |
>
> The probing baselines indeed provide a valuable reference. Verbal Confidence tends to be overconfident, resulting in only modest improvements in truthfulness. Conversely, Probing is more conservative; while it substantially increases truthfulness, it does so at the cost of significantly reduced accuracy (e.g., dropping from 28.27 to 18.27 on Qwen-7B). BARREL consistently outperforms these baselines, achieving a superior balance between accuracy, truthfulness, and reliability.
>
> We will include these results in the revised main figure and baseline section. As noted in [R9], we will also move the vanilla GRPO analysis to the main paper.
>
> [7] Tian, K., Mitchell, E., Zhou, A., Sharma, A., Rafailov, R., Yao, H., Finn, C., & Manning, C. (2023). Just Ask for Calibration: Strategies for Eliciting Calibrated Confidence Scores from Language Models Fine-Tuned with Human Feedback. EMNLP 2023.
>
> [8] Orgad, H., Toker, M., Gekhman, Z., Reichart, R., Szpektor, I., Kotek, H., & Belinkov, Y. (2025). Llms know more than they show: On the intrinsic representation of llm hallucinations. ICLR 2025.

---

> > ### Comment · Reviewer_2MeN · 2025-11-24
> > **Response to rebuttal**
> >
> > Thank you for the detailed rebuttal, I have raised my score. Few points/clarifications which should be addressed in the final version-
> >
> > **Reward Threshold**: I think the authors are mistaken when they say that the static threshold is not applicable because go GRPO. The RL algorithm and the underlying reward function are separate - every RL algorithm maximizes the reward function. In the reward function defined by the authors, it is optimal to say "idk" if the model's internal confidence is less than <0.5 as the expected reward is maximized by the abstain decision only in this case. This naturally means that the policy is tied to a fixed threshold level. If authors train with a different $r_s$ (for example 0.9), they will obtain a model that abstains significantly more. Now I personally think having a fixed $r_s$ is a bad thing. Anyone who wants to use a different refusal reward will have to retrain the model. Perhaps a way around this is to train model with a range of $r_s$, which is specified in the prompt. That is, the prompt contains the rejection reward $r_s$. This can yield a model which generalizes to multiple $r_s$. I think this should at least be acknowledged as a limitation or added to the discussion in some way.
> >
> > **Probing/Classifier Baselines**: How did authors select thresholds that yielded the best tradeoff? I guess this makes the baselines better, but just curious on why authors did not use 0.5 as the threshold (which is what the implicit threshold is for their reward function). I am also wondering if it would be possible to include calibration charts of these classifier/probes to understand where they are lacking.
> >
> > **Sampling Inconsistency**: Similarly, inconsistency in BARREL can be as high as 20%. This means that for 20% of the questions, the model does not know whether to abstain or not. I know that this is more consistent than the baselines, but it is still a high number in absolute sense. It should be acknowledged as a limitation or added to the discussion somewhere.
> >
> > **Knowledge Labeling Proxy**: I do not fully understand the author's arguments here. Say I ask the model a question where it is confused between two answers (A and B). As humans, we are often also confused between multiple options. As a trivial example, "I know the capital of the US is New York of Washington, but I can't remember which one". In this case, sometimes the LLM would output "New York" and sometimes "Washington". Under their framework, this would be classified as a question which is known to the model, which is not true (the model is genuinely confused between two options). I feel there are better proxies that can be constructed for SFT.

---

> ### Author Response · Authors · 2025-11-25
> **Response by Authors**
>
> We sincerely thank the reviewer for the prompt response. We are also grateful for your acknowledgement of our rebuttal and for raising the score. Regarding the points where we reached a consensus (including the revision of organization and presentation, moving the GRPO results from the Appendix to the main text, and adding new baselines), we have incorporated these changes into the latest manuscript and highlighted them in blue. Below is our response to the follow-up suggestions for the final version:
>
> **1. Reward Threshold:** We agree with the reviewer that training the model with a range of $r_s$ specified in the prompt could be a more flexible approach and represents a promising direction for future work. We have added a discussion regarding this in the Limitations section (Line 1457).
>
> **2. Probing/Classifier Baselines:** Our primary criterion for selection was achieving the best relative reliability metrics, as the model's confidence scores do not always reflect true probabilities. As shown by the calibration curves, the model's prediction confidence value tends to be overconfident. Since figures cannot be displayed here, we have included a discussion on this matter and the corresponding calibration curves in Appendix D: DETAILS ON VANILLA GRPO WITH OTHER ABSTENTION TECHNIQUES.
>
> **3. Sampling Inconsistency:** We have added relevant discussions in Appendix K: ANALYSIS OF SAMPLING VARIANCE AND STABILITY. Furthermore, we have incorporated this point into the Limitations section (Line 1449).
>
> **4. Knowledge Labeling Proxy:** We agree that the limitations in this scenario should be mentioned. While the original Limitations section mentioned that the sampling strategy—despite its widespread adoption—is prone to certain mislabeling issues, we have now strengthened the discussion specifically addressing the mislabeling cases pointed out by the reviewer (Line 1440).
>
> We thank the reviewer for the detailed and insightful questions and suggestions. We have uploaded a revised manuscript that incorporates the changes discussed in our rebuttal. Please let us know if there are any further areas requiring discussion or modification. We genuinely appreciate your time and effort.

---

### Official Review · Reviewer_cpNK · 2025-10-31

**Soundness:** 3
**Presentation:** 2
**Contribution:** 3
**Rating:** 6
**Confidence:** 4

**Summary:**

The paper identifies two pathological reasoning modes in LRMs—last-minute guessing and second-thought spiraling—and proposes BARREL, a two-stage training framework (SFT on curated reasoning traces + GRPO with a three-level reward: correct > truthful refusal > incorrect). Experiments on factual QA (TriviaQA, SciQ, NQ-Open) report large gains in a proposed Reliability metric and truthfulness, while preserving accuracy and math ability (MATH-500). The method emphasizes teaching models to admit ignorance and uses reward shaping to avoid penalizing refusals as harshly as wrong answers.

**Strengths:**

* Clear diagnosis of failure modes with concrete qualitative examples and a simple taxonomy (last-minute guessing vs. second-thought spiraling).
* Conceptually clean training recipe: SFT to seed refusal patterns + GRPO that explicitly rewards calibrated abstention.
* Consistent empirical improvements in reliability/truthfulness across multiple 7–8B backbones, with additional OOD refusal tests and pass@k analysis.
* Ablations that matter: effect of refusal reward magnitude; role of SFT vs. GRPO.

**Weaknesses:**

1. **Generalizability Issues in Reasoning Trace Construction**
The paper states that reasoning traces are constructed by prompting GPT-4 with prompts from Appendix F and BARREL reasoning examples to generate compliant reasoning patterns. However, this generation approach may lead to insufficient diversity in the resulting traces. Additionally, as a non-reasoning-focused model, GPT-4 may not be optimal for generating high-quality Long-CoT-style reasoning processes. A more effective alternative could involve **rewriting failed trajectories** produced by reasoning models using the BARREL framework—this would likely yield higher-quality datasets. In practice, model evaluations often generate a large number of failed cases characterized by *last-minute guessing* and *second-thought spiraling*; leveraging these failed cases via BARREL’s methodology would enable the creation of a more extensive and useful dataset.

2. **Unaddressed Response Length in Post-Training Evaluation**
The paper uses BARREL-generated datasets for SFT and GRPO training, and optimizes the model’s original Long-CoT trajectory generation process. This optimization may inadvertently impact the model’s response length—potentially reducing the analysis depth of certain reasoning paths. Notably, Figure 2 explicitly highlights response length differences between correct and incorrect answers in pre-trained models. Given this, it is necessary to **supplement the response length data** for the evaluation results in Table 1, alongside a corresponding analysis of how training affects response length and reasoning depth.

3. **Lack of Overthinking Analysis in Non-Factual Domains**
While the paper analyzes and optimizes overthinking in the factual domain, LLMs often exhibit severe overthinking in other domains as well. For example: (1) Mathematical reasoning, where overthinking occurs across multiple problem-solving approaches; (2) Instruction following, where overthinking arises from misinterpreting diverse task requirements. Intuitively, the BARREL framework could be applicable to these domains, but the paper provides no analysis or validation of such cross-domain applicability.

**Questions:**

1. While I understand the authors may have adopted the current data generation approach due to cost considerations, conducting a small-scale ablation study (as proposed in Weakness 1—i.e., comparing GPT-4-generated traces vs. BARREL-rewritten failed traces) would significantly enhance the paper’s validity and the practical significance of BARREL. Could the authors consider adding such an ablation study?

2. As noted in Weakness 2, response length is a critical indicator linked to overthinking (per Figure 2). Could the authors supplement the response length data for the evaluation results in Table 1, and provide an analysis of the underlying causes (e.g., how SFT/GRPO training modulates response length, and whether shorter/longer traces correlate with improved reliability)?

3. The paper focuses exclusively on overthinking in the factual domain. Given BARREL’s conceptual design (boundary-aware reasoning + calibrated reward), it may have broader applicability. Could the authors provide a preliminary analysis of BARREL’s suitability for addressing overthinking in other domains (e.g., mathematics, instruction following)—such as a pilot experiment or a discussion of potential adaptations to the framework?

---

> ### Author Response · Authors · 2025-11-20
> **Response by Authors (1/3)**
>
> We sincerely thank the reviewer for the detailed and insightful review. We especially appreciate the positive remarks on the clear motivation, method design and the comprehensive experiments and analyses. We provide our feedbacks as follows.
>
> > **[Q1]**: While I understand the authors may have adopted the current data generation approach due to cost considerations, conducting a small-scale ablation study (as proposed in Weakness 1—i.e., comparing GPT-4-generated traces vs. BARREL-rewritten failed traces) would significantly enhance the paper's validity and the practical significance of BARREL. Could the authors consider adding such an ablation study? (Weakness 1 and Question 1)
>
> **[R1]**: Thanks for the insightful and constructive suggestion regarding the generalizability of our reasoning trace construction. We fully agree that comparing our current GPT-4-based Constructing method with a Rewriting approach (revising failed model trajectories into BARREL traces) is a promising step for enhancing the validity and practical significance of the BARREL framework.
> As the reviewer noted, while computational budget constraints limited our initial scope, we have now performed the requested ablation study only on DeepSeek-R1-Distill-Llama-8B to provide initial evidence. The BARREL (Rewriting) method uses GPT-4 to rewrite model-generated reasoning traces into the BARREL format, whereas BARREL (Constructing) is our default pipeline (generating traces from scratch). The subsequent GRPO stage remains identical.
>
> | Method / Metric                      | Acc (Avg.) | Truth (Avg.) | Rel (Avg.) |
> |--------------------------------------|------------|--------------|------------|
> | ICL-IDK                              | 28.13      | 30.87        | 30.79      |
> | BARREL SFT only (Constructing)       | 31.87      | 49.77        | 46.56      |
> | BARREL Full process (Constructing)   | 40.70      | 70.40        | 61.58      |
> | BARREL SFT only (Rewriting)          | 27.03      | 44.13        | 41.21      |
> | BARREL Full process (Rewriting)      | 41.20      | 73.80        | 63.17      |
>
> For SFT stage, the Constructing approach performs better ($\sim 5$ points across metrics). This is expected for small models, as rewriting traces produced by a relatively weak 7B-scale model is inherently more challenging than generating high-quality BARREL-style reasoning from scratch. Thus, for small models, the constructive strategy is slightly more effective in the initial SFT stage.
> After applying the BARREL GRPO stage, the performance of the two methods becomes statistically comparable (with Rewriting slightly outperforming in reliability metrics). Crucially, this suggests that the GRPO stage effectively mitigates initial differences in SFT data quality, enabling the model to adapt and generalize well regardless of whether BARREL traces were constructed or rewritten.
>
> These results confirm that rewriting real reasoning traces into BARREL-style ones is also a viable and effective path for improving reasoning reliability. While our primary focus on smaller models led to the initial preference for the constructive method, we agree that rewriting-based training is likely even more practical and scalable for larger models by directly leveraging real-world failure cases.
>
> We will integrate this valuable ablation study into Section 4.3 (Analysis) and include the corresponding technical details in the Appendix of the final version.

---

> ### Author Response · Authors · 2025-11-20
> **Response by Authors (2/3)**
>
> > **[Q2]**:  As noted in Weakness 2, response length is a critical indicator linked to overthinking (per Figure 2). Could the authors supplement the response length data for the evaluation results in Table 1, and provide an analysis of the underlying causes (e.g., how SFT/GRPO training modulates response length, and whether shorter/longer traces correlate with improved reliability)? (Weakness 2 and Question 2)
>
> **[R2]**:Thanks for the reviewer's insightful question. To more comprehensively address the concern regarding response length and its potential connection to reasoning depth, we provide extended statistics corresponding to Table 1, including (1) the accuracy and reliability score, (2) the average number of thinking tokens for correct vs. incorrect samples and the Wrong/Correct ratio highlighted in Figure 2, and (3) the overall average response length. Results are shown in the table below.
>
> | Model | Accuracy | Reliability | Thinking Tokens on Correct Ones | Thinking Tokens on Wrong Ones | Wrong/Correct | Avg. Token |
> |---|---|---|---|---|---|---|
> | DeepSeek-R1-Distill-Llama-8B  | 28.13 | 28.33 | 421 | 561 | 1.33x  | 522  |
> | DeepSeek-R1-Distill-Llama-8B (BARREL SFT only) | 31.87 |46.56| 470  | 481 | 1.02x  | 476  |
> | DeepSeek-R1-Distill-Llama-8B (BARREL)| 40.70 | 61.58  | 442   | 458 | 1.04x  | 455  |
> | DeepSeek-R1-Distill-Qwen-7B|  18.07 |18.79| 362   | 484 | 1.34x  | 458  |
> | DeepSeek-R1-Distill-Qwen-7B (BARREL SFT only) | 20.53 | 35.48 | 473  | 506 | 1.07x  | 489  |
> | DeepSeek-R1-Distill-Qwen-7B (BARREL)  |28.27 | 53.12 | 429   | 440 | 1.03x  | 429  |
> | Qwen3-8B   | 41.97| 42.40 | 477   | 826 | 1.73x  | 676  |
> | Qwen3-8B (BARREL SFT only)| 37.67 | 50.56 | 478   | 485 | 1.01x  | 479  |
> | Qwen3-8B (BARREL) | 50.5 | 71.46 | 414   | 430 | 1.04x  | 433  |
>
> Overthinking phenomenon emphasized in Figure 2 indicates that incorrect trajectories are substantially longer than correct ones. From the table, we could observe that BARREL SFT stage consistently removes this asymmetry, reducing the Wrong/Correct ratio from 1.3×–1.7× to ~1.02×. This indicates that SFT avoids models to diverge into unnecessarily verbose chains—thereby directly mitigating the overthinking misbehavior. GRPO further reduces the average response length, while keeping the Wrong/Correct ratio near 1.0. This suggests that GRPO contributes a further generalization stage, reinforcing not only correct and truthful but also concise reasoning patterns. Importantly, the reduction in asymmetry does not come at the cost of reasoning depth. Crucially, as shown in Table 1 and this table, improvements in accuracy and reliability show that the shorter trajectories are not underthinking; instead, they reflect more concise and reliable reasoning shaped by the training objective.
>
> We also emphasize that absolute average response length is not a reliable standalone indicator of overthinking. In some cases (e.g., Qwen-7B), BARREL SFT yields slightly longer outputs than the base model, yet overall reliability and accuracy still increase. Our results collectively indicate that overthinking primarily manifests as the relative imbalance between correct and incorrect traces, rather than the absolute number of tokens generated.
>
> We will incorporate a concise paragraph summarizing these findings near line 342 and provide the full supplementary table and discussion in Appendix A of the final version.

---

> ### Author Response · Authors · 2025-11-20
> **Response by Authors (3/3)**
>
> > **[Q3]**:  The paper focuses exclusively on overthinking in the factual domain. Given BARREL's conceptual design (boundary-aware reasoning + calibrated reward), it may have broader applicability. Could the authors provide a preliminary analysis of BARREL's suitability for addressing overthinking in other domains (e.g., mathematics, instruction following)—such as a pilot experiment or a discussion of potential adaptations to the framework? (Weakness 3 and Question 3)
>
> **[R3]**: We thank the reviewer for this insightful question regarding domain generalization. To provide a precise answer, we clarify our scope: our work focuses on reliability (knowledge boundary awareness). We treat "overthinking" not as the definition of unreliability, but as its primary symptom within the factual domain.
>
> While our current experiments focus on factual QA, the BARREL framework follows a principled training recipe with broader applicability (as the reviewer mentioned in Strength 2): using SFT to seed refusal behavior and GRPO to introduce calibrated abstention incentives. This structure is compatible with mathematics, coding, and instruction following, though the specific implementation of "reliable reasoning" differs by domain:
>
> - In Factual QA (Current Work): Unreliability manifests as "overthinking" (hallucinating long rationales for unknown facts). Therefore, our method constructs training data specifically to curb this behavior, establishing a clear pattern for recognizing knowledge gaps.
>
> - In Other Domains (e.g., Math, Instruction Following): "Overthinking" is not the primary failure mode. For example, in tasks like IFEval, reasoning focuses on planning rather than fact retrieval. Our analysis shows minimal difference in thought chain length between correct and incorrect cases (e.g., 496 vs. 501 tokens for DeepSeek-R1-Distill-Llama-8B), suggesting length is not a boundary signal here.
>
> Adapting BARREL to these domains is an exciting open question. Different domains require modeling different boundary patterns:
>
> 1. For Instruction Following: Reliability is tied to constraint satisfaction and execution planning, not fact retrieval. A reliable reasoning pattern would involve simulating execution steps and detecting potential constraint violations before committing to an answer.
>
> 2. For Mathematics: Tasks involve complex logical reasoning chains. Here, mitigating unreliability requires improving sensitivity to intermediate reasoning errors. A reliable process would entail detecting when a logical step becomes uncertain, interrupting the flawed chain, and transitioning to an expression of uncertainty rather than completing a hallucinated derivation.
>
> Though reliable reasoning pattern for different domains differs, our BARREL training framework is a valuable first step to verify LRMs' ability to reasoning reliably and become more uncertain aware. BARREL’s framework can support these settings once the domain-specific reliable reasoning patterns are formalized and used to construct SFT refusal data, with GRPO rewards calibrated to these new signals.
>
> The above analysis and discussions will be included in the final version of the paper.

---

### Official Review · Reviewer_xT1G · 2025-11-01

**Soundness:** 3
**Presentation:** 4
**Contribution:** 3
**Rating:** 8
**Confidence:** 4

**Summary:**

This paper studies the problem of Large Reasoning Models rarely admitting ignorance and producing incorrect answers instead. In this work, the authors identify two pathological reasoning patterns, called Last-minute Guessing and Second-thought Spiraling, caused by overthinking.

To address these problems, the authors proposed a very clean framework, called BARREL. The frameworks clearly build SFT and RL data for these two patterns of wrong CoT. The results on TriviaQA, SciQ, and NQ-Open clearly show the effectiveness of the framework.

**Strengths:**

1. This paper provides a very clear motivation for fixing two pathological reasoning patterns in current Large Reasoning Models, which is a very important topic.
2. The writing of this paper is fluent and easy to follow.
3. The experiments and analysis in this paper are comprehensive to show the effectiveness of the proposed framework.

All in all, this is a solid paper with clear structure.

**Weaknesses:**

The only weaknesses I noticed are the definition of the metric "Reliability (Rel.)." The "ans." means the ratio of answered questions. So, why do we need to compute "ans.·Truth." and "(1−ans.)·Acc." and sum them together? What is the rationale behind this, and why do you think this is a more robust metric?

**Questions:**

See weakness

---

> ### Author Response · Authors · 2025-11-20
> **Response by Authors**
>
> We sincerely thank the reviewer for the detailed and constructive feedback, and for recognizing our work as a solid paper with a clear structure. We especially appreciate your positive remarks on (i) the clear motivation in addressing two pathological reasoning patterns in LRMs, (ii) the fluency and readability of the writing, and (iii) the comprehensive experiments and analyses. Below we respond to your question regarding the definition of the "Reliability (Rel.)” metric.
>
> **[Q1]**: The only weaknesses I noticed are the definition of the metric "Reliability (Rel.)." The "ans." means the ratio of answered questions. So, why do we need to compute "ans.·Truth." and "(1−ans.)·Acc." and sum them together? What is the rationale behind this, and why do you think this is a more robust metric?
>
> **[R1]**: Thanks for the thoughtful question. We adopt this formulation following the established practice in prior work [1, 2]. The central limitation of using truthfulness alone is that it can be trivially optimized by refusing to answer. As noted in line 288, a model that always replies "I don't know” would achieve perfect truthfulness while being completely unhelpful in practice. A more reliable evaluation therefore needs to jointly capture: (1) the risk of providing incorrect answers, and (2) the cost of being overly conservative.
>
> $\text{Reliability} = (\text{ans.}) \cdot \text{Truth} + (1-\text{ans.})\cdot \text{Acc.}$
>
> The reliability metric explicitly balances these two aspects by weighting performance on answered versus unanswered cases: (1) When the answer rate is low (small ans.), the metric is dominated by (1−ans.)⋅Acc. In this regime, a model that refuses everything but does so indiscriminately will not achieve high reliability. This encourages the model to be more helpful rather than excessively abstaining. (2) When the answer rate is high (large ans.), the term (ans.)⋅Truth dominates. In this case, the metric rewards models that remain truthful while answering many questions, and penalizes those that respond too aggressively with low truthfulness.
>
> Thus, even if two models have similar raw truthfulness on answered questions, the reliability score can distinguish whether a model is (1) responsibly conservative (refusing mainly when it should), or (2) simply avoiding engagement (refusing too broadly) or over-confident (answering too much with low truthfulness). This makes reliability a more faithful measure of overall practical robustness than truthfulness alone.
>
> [1] Xu, H., Zhu, Z., Zhang, S., Ma, D., Fan, S., Chen, L., & Yu, K. (2024). Rejection improves reliability: Training LLMs to refuse unknown questions using RL from knowledge feedback. COLM 2024
>
> [2] Kim, H. J., Kim, Y., Lee, S., & Kim, T. (2025). When to speak, when to abstain: Contrastive decoding with abstention. ACL 2025.

---

> ### Comment · Reviewer_xT1G · 2025-11-25
> **Reply from Reviewer xT1G**
>
> Thanks for your response. I will keep my positive score (8).

---

> > ### Author Response · Authors · 2025-11-25
> > **Response by Authors**
> >
> > We sincerely appreciate your timely and positive feedback and are pleased to know that our response addressed your concerns. Thank you for your thoughtful review and the time you invested in evaluating our work.

---

### Author Response · Authors · 2025-11-30
**Summary of Rebuttal**

Dear AC,

Acknowledging the significant workload ACs are facing, we have prepared a concise summary of our rebuttal progress to assist in your review. We hope this provides a clear and efficient overview of our paper. We sincerely appreciate the time and effort you are dedicating to managing this process.

Below is a summary of the discussion. **We have reached a consensus with xT1G and 2MeN, while cpNK has not yet responded.**

## **1. xT1G (Original: 8 $\rightarrow$ Current: 8; Status: Consensus Reached)**

The reviewer’s only concern regarding the reliability metric was fully addressed. xT1G confirmed the decision to maintain the positive score (8) on Nov 25, 2025 AOE.

## **2. 2MeN (Original: 4 $\rightarrow$ Current: 6; Status: Consensus Reached)**

**Reviewer 2MeN raised the score to 6** following our detailed rebuttal and provided constructive suggestions for the final version (Nov 24, 2025 AOE), which we have fully incorporated. We addressed the reviewer's concerns across four major categories:

- **Part 1**: Metrics, Settings, and Assumptions

  - **Addressed**: We clarified the design of our reward mechanism [Q3] and justified the Reliability Metric [Q2] and Knowledge Labeling Proxy [Q1] using established literature.

  - **Final Action**: As requested in the final round, we explicitly discussed the limitations of the proxy definition and the flexibility of reward design in the Limitations section of the revised manuscript.

- **Part 2**: Interpretation and Reliability of Analysis

  - **Addressed**: We resolved [Q4] (SFT vs. ICL capabilities) and [Q5] (Pass@k vs. Pass@1 trade-off). The reviewer confirmed that no further issues remain regarding these points.

- **Part 3**: Organization and Presentation

  - **Addressed**: We adopted the reviewer's suggestions regarding structural improvements.

  - **Final Action**: We moved the GRPO-related results from the Appendix to the Main Text [Q9], added the average abstain rate [Q7], moved specific analyses to the Appendix [Q8], and integrated additional related work on RL-based uncertainty [Q6].

- **Part 4**: Additional Experimental Analyses

  - **Addressed**: We added an analysis on sampling variance [Q10] and implemented two requested baselines [Q11]: GRPO with Verbal Confidence and Probing. The reviewer acknowledged these results and suggested adding the variance analysis to the limitations and including the new baselines.

  - **Final Action**:  As requested in the final round, we included the variance analysis in Appendix K and the Limitations section to address sampling stability. We also included the new baselines in the main text and placed the further suggested analysis in the Appendix.

## **3. cpNK (Original: 6; Status: Not Replied)**

Reviewer cpNK’s primary concerns center on the scalability and generalizability of our method. We have addressed these issues in detail, supported by experimental results. A summary is provided below:

- [Q1] (Constructing new traces vs. Rewriting failed ones): We demonstrated through ablation experiments that the rewriting strategy is also highly effective, verifying the scalability of our framework.

- [Q2] (Response length statistics & "overthinking"): We showed that BARREL mitigates factual overthinking phenomenon and proved that the resulting concise traces reflect reliable reasoning rather than underthinking.

- [Q3] (Applicability to other domains): We clarified that the core BARREL training strategy is universally applicable, provided that domain-specific reliable reasoning patterns are defined.

We have incorporate these additional experiments and analysis in the main text (Page 10), Appendix P and Appendix R.

## **Summary**

We appreciate the recognition of the **clear motivation and precise problem diagnosis** (xT1G, cpNK, 2MeN), **the clean methodology and comprehensive experiments** (xT1G, cpNK, 2MeN), and **the fluent writing and structure** (xT1G, 2MeN). We are pleased to have resolved the concerns of reviewers xT1G and 2MeN and reached a consensus during the discussion period. Furthermore, we believe our response to reviewer cpNK effectively addresses their related concerns.

We sincerely appreciate the time and effort dedicated by the AC and all reviewers, and we hope this summary aids in your final evaluation. If you have any further inquiries, we are happy to engage in further discussion to address them.

---

### Meta-Review · Area_Chair_tRv5 · 2026-01-07

**Summary:**

This paper propose a two-stage training framework called BARREL to improve the LRM's factual knowledge through SFT and GRPO. Experiments demonstrate large gains on truthfulness on factual QA benchmarks while preserving performance of the model. Overall, the reviewers find this paper's motivation and the proposed method is clear with comprehensive experiments. Most reviewers are positive about the paper after rebuttal and hence a recommendation of accept.

**Reviewer Concerns:**

* Reviewer xT1G's concern on reliability metric is addressed
* Reviewer 2MeN's concern on metrics, reliability analysis, presentation, and additional experiment analysis are addressed
* Reviewer cpNK's concern on scalability and generalization is addressed

**Reviewer Scores:**

* Reviewer xT1G stated to remain score of 8
* Reviewer 2MeN stated to increase score to 6
* Reviewer cpNK likely will remain score of 6

---

### Decision · Program_Chairs · 2026-01-26

Accept (Poster)